# Current Trends in Cancer Immunotherapy

**DOI:** 10.3390/biomedicines8120621

**Published:** 2020-12-17

**Authors:** Ivan Y. Filin, Valeriya V. Solovyeva, Kristina V. Kitaeva, Catrin S. Rutland, Albert A. Rizvanov

**Affiliations:** 1Institute of Fundamental Medicine and Biology, Kazan Federal University, 420008 Kazan, Russia; IvYFilin@kpfu.ru (I.Y.F.); VaVSoloveva@kpfu.ru (V.V.S.); KrVKitaeva@kpfu.ru (K.V.K.); 2Faculty of Medicine and Health Science, University of Nottingham, Nottingham NG7 2QL, UK; catrin.rutland@nottingham.ac.uk; 3Republic Clinical Hospital, 420064 Kazan, Russia

**Keywords:** immunotherapy, cancer, immune checkpoint inhibitors, cytokine therapy, vaccines, oncolytic viruses, CAR T-cell therapy

## Abstract

The search for an effective drug to treat oncological diseases, which have become the main scourge of mankind, has generated a lot of methods for studying this affliction. It has also become a serious challenge for scientists and clinicians who have needed to invent new ways of overcoming the problems encountered during treatments, and have also made important discoveries pertaining to fundamental issues relating to the emergence and development of malignant neoplasms. Understanding the basics of the human immune system interactions with tumor cells has enabled new cancer immunotherapy strategies. The initial successes observed in immunotherapy led to new methods of treating cancer and attracted the attention of the scientific and clinical communities due to the prospects of these methods. Nevertheless, there are still many problems that prevent immunotherapy from calling itself an effective drug in the fight against malignant neoplasms. This review examines the current state of affairs for each immunotherapy method, the effectiveness of the strategies under study, as well as possible ways to overcome the problems that have arisen and increase their therapeutic potentials.

## 1. Introduction

In 1909, Paul Ehrlich predicted that the immune system normally prevents the formation of carcinomas from various origins. The first demonstration of a specific immune system response was made almost half a century later in 1953 by Gross [1]. Unfortunately, malignant neoplasms may evade such immune responses. It is known that malignant neoplasms downregulate the molecules of the major histocompatibility complex (MHC)-I, thereby preventing recognition of tumor cells by cytotoxic T-lymphocytes (CTLs). However, the immune system, in turn, is able to destroy cells that do not express, or insufficiently express, MHC-I molecules on their surface using natural killer cells (NK-cells) [2]. However, tumor cells can also protect themselves from NK-cell lysis by expressing non-classic human leukocyte antigen (HLA)-G molecules on their surface [3]. Additionally, tumor cells can trigger angiogenesis [4], and can also recruit T-regulatory cells with immunosuppressive properties via chemical signals [3]. Thus, malignant neoplasms may evade the host immune response, creating a tumor microenvironment. Gavin Dunn and Robert Schreiber developed the concept of “cancer immunoediting” in three phases. In the first phase, tumor cells are eliminated by cells of the immune system (NK cells, CD4^+^ and CD8^+^ T-lymphocytes) [5]. In the second phase, there is an equilibrium between tumor cells and cells of the immune system. In the third phase, the immune system is unable to cope with the tumor, which has an impressive immunosuppressive effect; therefore, the phase ends with the appearance of a clinically detectable tumor [5]. Nowadays, the priority task is to create an effective immunotherapeutic method with minimal toxicity to overcome the immunosuppressive activity of the tumor cells and enhance targeted elimination of the tumor by the immune system host cells. Most immunotherapeutic methods, for example monoclonal antibodies, target specific antigens on the tumor cell surface to produce effective and accurate actions, and some methods, like dendritic vaccines, use these antigens to enhance the immunostimulating and immunomodulatory immune system activity. There are some types of antigens located on the surface of tumor cells that can induce a specific immune response. Such antigens were shown by Richmond Prehn and Joan Mine in their murine experiments in 1957 [6]. Subsequently, the so-called tumor-associated antigens (TAAs) were discovered. These include molecules that are expressed on the surface of cells in a prevailing amount, or in a state different from that observed in normal cells. Other tumor biomarkers include tumor-specific antigens (TSAs), which are fragments of novel peptides that are presented by MHC-I at the cell surface. The first TSAs were discovered in 1991 in human melanoma cells, which are encoded by the melanoma-associated antigen (MAGE) gene family [7]. There are also neoantigens that are the result of somatic mutations and are specific to each patient, thus differing from wild-type antigens. These antigens are used in immunotherapy methods like a target for the recognition and subsequent elimination of tumor cells. Therefore, cancer immunotherapy aims to use the memory and specificity of the immune system to effectively eliminate malignant neoplasms for long periods of time and with minimal toxicity. Immunotherapy methods are also aimed at stimulating the body’s own immune system in order to fight off the tumor. Immunotherapeutic methods presently include cytokine therapy, monoclonal antibodies, oncolytic viruses, prophylactic and therapeutic vaccines, and chimeric antigen receptor (CAR) T-cell therapy.

## 2. Cancer Immunity

The relationship between a tumor and the human immune system is dynamic and complex. The process of detecting and eliminating tumor cells in the body is a stepwise process, and both cellular and humoral immunity are involved (Figure 1). Formation of tumor cells in the human body is problematic as both have multifactorial profiles, but the body’s immune cells constantly patrol its surroundings and when neoantigens created as a result of oncogenesis are present, dendritic cells (DCs) take them up by phagocytosis or endocytosis. Pro-inflammatory cytokines and other factors direct the immune response, whereby DCs migrate to the lymph nodes to present the captured antigens on MHC-I and MHC-II molecules to T-lymphocytes. Then, there are priming, activation and differentiation of the T-cell response, representing the ratio of T-effector and T-regulatory cells (Treg). Thereafter, T-helpers (Th) 1 produce interleukin (IL)-2 for the clonal expansion of activated cytotoxic T-cells, which then penetrate into the tumor bed, and recognize and bind to cancer cells via the T-cell receptor (TCR) and MHC-I [8].

Furthermore, Th2 can induce a humoral immune response which leads to the activation and proliferation of B-cells. Moreover, soluble antigens can be acquired by B-cells directly from the lymph or they may encounter tumor neoantigens on the surfaces of DCs and macrophages. B-cells can also present tumor antigens to naïve CD4^+^ T-cells [9]. B-cells help with the destruction of tumor cells through antibody-dependent cytotoxicity mechanisms secreting antibodies that “mark” tumor cells. In addition, B-cells can produce cytokines that coordinate other immune cells; in particular, they can enhance cytotoxic T-cell activity. However, despite these protective characteristics, they also have destructive ones. Antibodies directed towards tumor-associated antigens can just block access to cellular effect factors, thereby forming a tumor defense. However, these antibodies have also attracted interest as potential biomarkers for early diagnosis of different types of cancer [10]. T-cells and NK-cells play key roles in the elimination of tumor cells. Moreover, the subset of T-cells that are involved in this procedure is large. CD8^+^ CTLs are aimed at directly eliminating tumor cells. Th1 produces the pro-inflammatory cytokines interferon-γ (IFN-γ) and IL-2. Th2 helps B-cells and also produces anti-inflammatory cytokines IL-4, IL-5, and IL-13, whilst Th17 is involved in antimicrobial immunity and produces pro-inflammatory cytokines including IL-17A, IL-17F, and IL-22. On the other hand, Treg produces anti-inflammatory cytokines IL-10 and transforming growth factor β (TGF-β), and also has an immunosuppressive effect on immune cells upon contact. There is also a small subset of Th9 secreting IL-9, which affects Th17 differentiation and Treg function. If tumor cells do not have MHC-I molecules, but at the same time express stress molecules on their surface, which the NK-cells can recognize, and they will then destroy the tumor cells without preliminary differentiation. The release of even more antigens after the destruction of the tumor cell stimulates immune cells, increasing the response breadth and depth [11,12].

Despite their role in destroying tumor cells and suppressing their growth, immune cells and the immune system play a dual role in the development of cancer. Immune cells can also create favorable conditions within the tumor microenvironment (TME), in particular, Treg have an immunosuppressive effect on DCs and CTLs by anti-inflammatory cytokines (IL-10 and TGF-β). Moreover, these cytokines force CD4+ T-cells to differentiate into Treg, which have suppressive properties for both Th1 and Th2 [3]. Therefore, various factors within the TME can suppress effector cells, which can promote tumor growth [8]. In addition, tumor cells can hide their TAAs, ensuring that the T-cells are unable to recognize them (Figure 1). Under these conditions, antitumor immunotherapy should be aimed at enhancing immune responses and preventing immunosuppression.

## 3. Cytokine Therapy

Cytokines are polypeptides or glycoproteins produced by activated cells of the hematopoietic or immune system, which mediate growth and differentiation signals, as well as inflammatory or anti-inflammatory signals to other cells. Cytokine therapy, one of the most promising methods of immunotherapy, was formed due to the presence of proapoptotic or antiproliferative activity in cytokines, as well as the possible stimulation of cytotoxic activity in immune cells. The antitumor activity of cytokines was first described by Ion Gresser and Chantal Bourali using the example of recombinant murine IFN-α in 1970 [13]. Scientists carried out daily intraperitoneal injections of IFN-α into BALB/c, C57BL/6 and DBA/2 mice, inoculated intraperitoneally with several thousand Ehrlich ascites (EA), RC19, EL4, L1210, and E♂G2 tumor cells. The most pronounced antitumor effects were observed in the BALB/c line inoculated with EA cells. Thereafter, several fundamental conclusions were made: treatment of mice with IFN-α and without EA inoculation was ineffective; subcutaneous administration was less effective than intraperitoneal administration; the increased survival of mice with effective treatment was directly related to effective tumor inhibition and inversely proportional to the amount of EA inoculated; phagocytosis of tumor cells by macrophages was observed only on smears obtained from abdominal cavities treated with interferon; mice that survived inoculation of EA cells, which were injected with interferon, exhibited increased resistance to repeated inoculation of EA cells [13]. In subsequent years, this discovery gave rise to a large number of preclinical and clinical trials using recombinant cytokines for antitumor immunotherapy. However, the high expectations of scientists were not justified, due to the limitations that arose in clinical trials, such as the short half-life of most cytokines, which leads to low therapeutic effects which necessitated use of higher drug doses, which led to high levels of toxicity with concomitant side effects [8]. Only two cytokines, IL-2 and IFN-α, have been approved so far by the Food and Drug Administration (FDA), and have shown only moderate efficacy in studies [14]. These cytokines have demonstrated for the first time that immunotherapy can affect tumor development within the body and give a definite antitumor immune response, despite high toxicity levels [14,15,16].

Recombinant IFN-α is used in high doses for the treatment of melanoma, leukemia and Kaposi’s sarcoma [17,18,19]. The main side effects observed with IFN-α therapy have been fever, chills, flu-like symptoms, and more serious symptoms including hepatic dysfunction, thyroid disorders, depression and anorexia. IntronA is the most famous drug based on recombinant IFN-α, which was manufactured by MSD Pharmaceuticals until 2019. Various modifications of IFN-α appeared to be aimed at increasing the half-life, as well as increasing the immunostimulating properties, thus minimizing its toxicity [20,21]. However, the emergence of new, safer and more effective methods of immunotherapy has reduced the number of clinical trials with this protein.

IL-2 is a key cytokine that stimulates the proliferation of NK cells and T-lymphocytes; therefore, it is actively used both for adoptive cell therapy using T-lymphocytes and for direct administration to patients [14]. Recombinant IL-2 is the main active ingredient in Proleukin. This drug is approved for the treatment of metastatic melanoma and metastatic kidney cancer [22,23]. However, it also causes frequent side effects of grade 3 and 4, including: anemia, cardiac arrhythmia, fever, hypotension, metabolic acidosis, nausea, thrombocytopenia and organ failure, including hepatic and renal failure. Recently, second generation therapies based on IL-2 have been developed, which should increase the half-life, for example, by directed cytokine pegylation or by chimerization of the cytokine with antibodies that target the protein to the TME, and should increase pharmacodynamic properties through biotechnological methods [24,25]. Several compounds have been able to achieve clinical trial status. For example, NKTR-214 is a second generation recombinant IL-2 containing polyethylene glycol (PEG) molecules designed to prevent binding to IL-2Rα/CD25, targeting IL-2Rβ/CD122, which is found in certain immune cells (CD8^+^ T-lymphocytes and NK cells). IL-2 has high affinity binding to the IL-2Rα/CD25 receptor, which is highly expressed on Tregs, which can reduce the bioavailability of the cytokine for T-lymphocytes and NK cells, thereby reducing the antitumor effect. This modified cytokine is being actively tested in clinical trials in combination with immune checkpoint inhibitors (ICIs) such as pembrolizumab (NCT03138889), nivolumab and ipilimumab (NCT03282344, NCT03435640, NCT02983045).

In addition to the aforementioned immune mediators in the treatment of oncological diseases, the pharmacokinetic and pharmacodynamic properties of other cytokines, such as IL-10, IL-12, IL-15, IL-21, and granulocyte-macrophage colony-stimulating factor (GM-CSF), have been tested. Some of them are already being used in preclinical and clinical trials with varying degrees of success, mainly in combination with various drugs based on monoclonal antibodies or with adoptive T-lymphocytes [26,27,28,29]. Moreover, there are alternative strategies aimed at directly introducing vectors encoding a cytokine into TME using electroporation, adenoviruses, or oncolytic viruses. In one such study, in a phase I clinical trial, patients with metastatic melanoma were injected with plasmids encoding the IL-12 gene via electroporation on days 1, 5 and 8 over a 39-day cycle. Patients received electroporation into metastatic melanoma lesions. The biopsy showed a proportional increase in the level of the IL-12 protein with an increase in the administered dose of the plasmid with IL-12, as well as severe tumor necrosis and lymphocytic infiltration. Herewith, 8 out of 19 patients showed disease stabilization or partial response [30]. Another study evaluated the safety and biological effect of intratumoral injection of an adenoviral vector encoding the IL-12 gene. Patients were injected with between 2.5 × 10^10^ and 3 × 10^12^ viral particles in seven groups of patients with terminal stage of malignant neoplasms of the gastrointestinal tract (GIT). Disease stabilization was noted in 6 of the 21 patients. Thus, it has been shown that this intratumoral injection of an adenoviral vector encoding the IL-12 gene is safe, but has only moderate antitumor effects [31].

All in all, the development of cytokine therapy in the future has been reduced to the following options: to obtain effective combinations with other immunotherapy methods that have already received FDA approval, such as monoclonal antibodies against programmed cell death (PD-1) and its ligand (PD-L1) [32], cytotoxic T-lymphocyte associated protein 4 (CTLA-4) [33] and CAR T-cell therapy [34]; to improve the pharmacokinetics of the investigated mediators, namely modification by conjugation with PEG [35], construction of a fusion protein with apolipoprotein A-I (ApoA-I) [21,36], antibodies [37], Fc domain [38]; as well as the improvement of the pharmacodynamic properties associated with a decrease in binding to cytokine receptors that do not trigger the intracellular signaling cascade [24].

## 4. Monoclonal Antibodies

The development of monoclonal antibodies (mAbs) against tumor cells began in the 1970s [39]. Among therapeutic monoclonal antibodies, various types are widely used: human (adalimumab); humanized (trastuzumab), 90–95% human; chimeric (rituximab), 60–70% human; mouse (muromonab). The main idea was to target mAbs to TAAs and kill tumor cells. Destruction of target cells by mAbs can be achieved in several ways, such as direct antibody action (blockade of receptors or delivery of the target toxic agent), immune-mediated cell killing, specific antibody action on the vascular system and TME [40]. The first successes in the clinic were associated with the latter mechanism [41]. Bevacizumab, an FDA-approved angiogenesis inhibitor, is indicated for the treatment of metastatic colorectal cancer (mCRC) [42], non-small-cell lung cancer (NSCLC) [43], metastatic breast cancer (mBC) [44], glioblastoma multiforme (GBM) [45], renal cell carcinoma (RCC) [46], ovarian cancer (OC) [47] and cervical cancer (CC) [48].

Antibodies targeting TAAs are also a promising area of research. For example, the drug trastuzumab for the treatment of breast cancer [49,50], which blocks overexpression of human epidermal growth factor receptor 2 (HER2), which sends a signal for cell growth. Pertuzumab is another recombinant anti-HER2 humanized monoclonal antibody [51]. The difference is that trastuzumab and pertuzumab bind to different domains of HER2, which gives a synergistic effect [52]. The Clinical Evaluation of Pertuzumab and Trastuzumab (CLEOPATRA) study showed the efficacy of pertuzumab and trastuzumab in combination with docetaxel compared with a combination of placebo, trastuzumab and docetaxel in the treatment of metastatic breast cancer without increasing cardiac toxicity (NCT00567190). The objective response rate (ORR) was higher in the pertuzumab group. Complete response (CR) was observed in 14 control group patients and 19 patients in the pertuzumab group. Treatment also gave a partial response (PR) for 219 patients in the first group and 256 patients in the second group [53]. There is also ongoing research on adjuvant therapy in patients with HER2-positive breast cancer (Adjuvant Pertuzumab and Herceptin in Initial Therapy in Breast Cancer (APHINITY)) (NCT01358877). The HerMES observational study evaluated the safety and efficacy of trastuzumab in patients with HER2-positive gastric cancer or gastroesophageal junction (GEJ). The median overall survival (OS) was 14.1 months, and the median progression-free survival (PFS) was 7.9 months, with ORR at 43.4%. This study confirmed the positive results of the main clinical trial—a study of Herceptin (trastuzumab) in combination with chemotherapy compared with chemotherapy alone in patients with HER2-positive advanced gastric cancer (ToGA study) in Germany [54].

In addition to solid tumors, a lot of attention is also paid to hematological malignancies. Hematological malignancies, unlike solid tumors, begin in the blood-forming tissue of the bone marrow and lymphoid cells. Hematologic B-cell tumors represent a large heterogeneous group of lymphoproliferative disorders, including diseases such as follicular lymphoma (FL), chronic lymphocytic leukemia (CLL), mantle cell lymphoma (MCL), diffuse large B-cell lymphoma (DLBCL) and others [55,56]. B-cell transmembrane protein (CD20) was chosen for targeted therapy as it is expressed on most B-cells, including malignant B-cells [57]. The standard treatment among hematologic malignancies is rituximab, which is a chimeric anti-CD20 mAb for the treatment of non-Hodgkin’s lymphoma, CLL, rheumatoid arthritis, Wegener’s granulomatosis and microscopic polyangiitis. Clinical trials have shown that this drug not only prolongs PFS but also increases OS in patients with lymphoma [57]. However, development of resistance in rituximab-treated patients was observed [58]. Mechanisms of resistance were explained by trogocytosis of mAb-CD20 complexes [59] and by internalization of rituximab from the surface of the B-cells malignancies [60]. Obinutuzumab is a glycoengineered, humanized anti-CD20 mAb (type II) that has been developed to increase activity by enhancing binding affinity to the FcγRIII receptor on immune effector cells and due to the ability of enhancing direct cell death and antibody-dependent cell-mediated cytotoxicity/antibody-dependent cellular phagocytosis because of the modified elbow-hinge amino acid sequence [61].

Currently, there are 113 mAbs approved by the FDA, among which 48 mAbs, including biosimilars, are for the treatment of solid tumors and hematological malignancies. There are also mAbs which are chemically conjugated to cytotoxic drugs or radioactive isotopes for targeted delivery. For example, brentuximab vedotin is used to treat Hodgkin’s lymphoma and anaplastic large cell lymphoma (ALCL). Brentuximab vedotin consists of a human chimeric immunoglobulin G1 targeting CD30, which is linked to the agent monomethylauristatin E (MMAE) via a protease-cleavable linker [62]. This drug has shown its safety and efficacy in many clinical trials, and is being moved to earlier lines of therapy in ongoing trials [63]. Other antibody-drug conjugates (ADCs) that were approved by FDA and European Medicines Agency (EMA) are gemtuzumab ozogamicin for treating CD33-positive acute myeloid leukemia in combination and as a single-agent therapy [64,65]; trastuzumab emtansine for treating HER2-positive breast cancer for patients who previously received trastuzumab and a taxane [66,67]; inotuzumab ozogamicin for treating relapsed or refractory CD22-positive B-cell precursor ALL in combination and as a single agent [68]. There are also several ADCs that are under development and are being clinically tested such as Glembatumumab vedotin, Sacituzumab govitecan, Anetumab ravtansine, Coltuximab ravtansine, Trastuzumab deruxtecan and GSK2857916 [69].

mAbs fused with immunomodulatory antibodies are also being investigated to create bispecific antibodies for immuno-mediated destruction of tumor cells. Drugs such as blinatumomab, bispecific mAbs that help CD3^+^ T-cells recognize and destroy CD19^+^ cells in acute lymphoblastic leukemia (ALL). In a Phase III clinical trial, 405 ALL patients were randomly assigned to blinatumomab (271 patients) or chemotherapy (134 patients). Median OS was 7.7 months in the blinatumomab group and 4 months in the chemotherapy group (HR for death with blinatumomab vs. chemotherapy, 0.71; 95% CI, 0.55 to 0.93; *p* = 0.01). Complete remission with complete hematological recovery was observed in 33.6% of patients, and remission with complete, partial or incomplete hematological recovery was observed in 43.9% of patients in the blinatumomab group. This was undoubtedly higher when compared to the rates in the chemotherapy group—15.7% and 24.6%, respectively. Additionally, mean duration of remission among these patients was 7.3 months (95% CI, 5.8 to 9.9) in the blinatumomab group and 4.6 months (95% CI, 1.8 to 19.0) in the chemotherapy group. However, side effects of grade 3 or higher were observed in 87% of patients in the blinatumomab group and in 92% of patients in the chemotherapy group [70].

With significant success in combination with chemotherapy, mAbs also have a large number of side effects. Common side effects include chills, fever, severe weakness, headache, nausea, vomiting, diarrhea, a drop in blood pressure, skin rash, and certain problems may occur in patients prone to allergies. The most frequent side effects are infusion reactions. Such side effects can occur due to allergic reactions to foreign proteins or as a result of cytokine release. All described infusion reactions were observed in patients in the first infusions of rituximab; however, they varied in severity and included different symptoms [71]. Moreover, specific side effects may occur when mAbs targeting an antigen are used. Treatment with bevacizumab can cause gastrointestinal perforation, hypertension, bleeding, nausea, diarrhea, thromboembolic complications, and less commonly skin ulcers [72,73], in addition to bowel ischaemia and haemorrhage [74]. Conjugated mAbs can exhibit higher toxicity due to conjugated substances—toxic chemotherapy drugs or radionuclides. For example, in patients who have been treated with brentuximab vedotin, the most common side effects were peripheral sensory neuropathy (PSN), neutropenia, fatigue, nausea, anaemia, upper respiratory tract infection (URTI), diarrhoea, thrombocytopaenia and coughing [62]. More severe side effects are related to bispecific antibodies. Side effects of the central nervous system (CNS) such as encephalopathy, aphasia, tremor, disorientation, and seizure are observed in patients treated with blinatumomab [75]. In one study, 52% of patients in original cohort and 42% in the additional evaluation cohort had neurologic events (NEs). Moreover, 4 patients had grade 4 NEs (encephalopathy, ataxia, seizures and febrile delirium), and two of them were fatal after treatment had stopped [76]. The main reason for such side effects can be explained by adherence of blinatumomab-activated T-cells to the endothelium, so that activated T-cells transmigrate across the blood brain barrier and enter the CNS where they induce a T-cell mediated toxic inflammation of the CNS [75].

Some patients who have been treated with mAbs then show resistance to this therapy. It has been noticed that a high percentage of Treg and high serum lactate dehydrogenase levels are related to lack of response to blinatumomab. It is supposed that activation of Treg by blinatumomab, and further IL-10 production, results in suppression of T-cell proliferation [77]. In addition lack of response can be as a consequence of the formation antibody-drug antibodies (ADAs). Furthermore, patients on infliximab who have developed ADAs also developed ADAs to adalimumab [78]. Thus, it is necessary to calculate the dose of the drug to minimize the likelihood of ADAs development.

### Immune Checkpoint Inhibitors

An effective antitumor response in antibody therapy has been demonstrated by immunomodulatory antibodies targeting inhibitory T-cell receptors. James P. Allison and Tasuku Honjo were awarded the Nobel Prize in Physiology or Medicine “for their discovery of cancer therapy by suppressing negative immune regulation” in 2018. Immune checkpoint inhibitors (ICIs) are monoclonal antibodies that inhibit factors that suppress T-cell function, thereby activating the immune system. ICIs are an important method of cancer immunotherapy and are aimed at blocking several targets that interfere with the proliferation of T-lymphocytes, such as cytotoxic T-lymphocyte-associated protein 4 (CTLA-4) [79], programmed cell death 1 (PD-1) [80], programmed cell death ligand 1 (PD-L1) [81] (Figure 2).

There are several drugs that have been approved by the FDA for treating a wide variety of tumors, including melanoma [82,83], NSCLC [84], Hodgkin’s lymphoma [85], hepatocellular cancer (HCC) [86], renal cell carcinoma (RCC) [87] and others. Drugs such as ipilimumab, pembrolizumab, nivolumab, durvalumab have shown positive results in numerous clinical trials (Table 1). Nivolumab and dacarbazine were compared in patients with previously untreated BRAF wild-type advanced melanoma. At a minimum follow-up of 38.4 months among 210 participants (nivolumab group) and 208 participants (dacarbazine group), the median OS was 37.5 months and 11.2 months, respectively. CR and PR were 19.0% and 23.8% in the nivolumab group compared with 1.4% and 13.0% in the in the dacarbazine group [88]. Nivolumab in combination with ipilimumab showed better results in patients with advanced melanoma than nivolumab alone or ipilimumab alone. Overall survival at 5 years was 52% in the nivolumab-plus-ipilimumab group and 44% in the nivolumab group, as compared with 26% in the ipilimumab group [89]. With pembrolizumab plus chemotherapy for NSCLC, the median OS was 15.9 months, compared with chemotherapy alone, where the median OS was 11.3 months [90]. In Phase I/II clinical trials, the efficacy of durvalumab in patients with NSCLC was assessed, having previously determined the level of PD-L1 expression in tumor cells, considering the result to be positive at a PD-L1 expression level of ≥25%. ORR was 15.3% of the total number of patients, specifically 21.8% in patients with PD-L1 expression ≥25% and 6.4% in patients with PD-L1 expression <25%. The median OS of the total number of patients was 12.4 months, which broke down to 16.4 months for PD-L1 ≥25% and 7.6 months for PD-L1 <25%. The median PFS overall was 1.7 months, with 2.6 and 1.4 months in patients with PD-L1 expression ≥25% and <25%, respectively. Moreover, side effects were observed in 10.2% of patients [91]. The efficacy of pembrolizumab in patients with relapsed or refractory Hodgkin lymphoma was also analyzed. With a median follow-up of 27.6 months, ORR was 71.9%, the CR was 27.6%, and the PR rate was 44.3% [92]. In another study, pembrolizumab in patients with relapsed/refractory classical Hodgkin lymphoma after autologous stem cell transplantation showed promising results. The PFS at 18 months for the 28 evaluable patients was 82%. The 18-month overall survival was 100% [93]. Nivolumab for newly diagnosed advanced-stage classic Hodgkin lymphoma was associated with promising efficacy. The ORR was 84%, 67% achieve complete remission. With a minimum follow-up of 9.4 months, the PFS was 92% [94].

ICIs block signaling pathways that regulate the immune system, thereby activating it, but at the same time, they induce inflammation, contributing to the development of immune-related adverse events (irAEs). During therapy, irAEs of various natures and genesis can be observed, including several organs or systems, for example: skin diseases (itching, rash, eczema, vitiligo) with 30–50% frequency; thyroid dysfunction (hypothyroidism, hyperthyroidism and thyroiditis) in 6–20% of patients; hypophysitis in 1–7% of patients; gastrointestinal diseases like diarrhea occurs in one third of patients or colitis in 8–22%; liver disease (hepatitis) with 1–4% frequency, but it reached 17.6%; lung diseases such as pneumonitis with 5–10% frequency, and, the most common, dyspnoea (53%) and cough (35%); rare endocrine adverse events such as primary adrenal insufficiency (0.7%) and insulin-deficient diabetes (0.2%), and others [95]. The risk of most iAEs is dose-dependent. These findings are supported by numerous randomized clinical trials (RCTs), in which irAEs were present in 54–96% of patients with progressive melanoma [96]. These adverse events can lead to serious consequences until death. To avoid such consequences, it is necessary to choose the safest dosage and frequency of administration of ICIs to patients, enabling the maximum effects with minimal toxicity to the body. In one study, researchers conducted a meta-analysis of RCTs for the treatment of progressive melanoma and found that using nivolumab, 3 mg/kg (3 milligrams of drug per kilogram of body weight) every 2 weeks, and pembrolizumab, 2 mg/kg every 3 weeks and 10 mg/kg every 3 weeks, the risk of serious irAEs is low [96]. In another article, researchers analyzed NSCLC patients who developed autoimmune encephalitis after using ICIs. This is one of the rare but serious complications of ICIs therapy and is characterized by a disorder of the nervous system. Researchers observed that patients who received corticosteroids after the onset of the disease had faster regression of symptoms without any complications [97]. It is assumed that the development of irAEs is commonly associated with inflammation due to pro-inflammatory cytokines and autoimmune reactions due to the cross-reactivity of T-cells. Such cross-reactivity may occur because of the antigenic resemblance between tumor cells and host cells [98]. It is hard to define the frequency and severity of irAEs because it depends on a variety factors such as the dose of ICIs, their mechanism of action and combination with other therapeutic agents, and other features of individual patients (such as underlying autoimmune disease, organ or hematopoietic stem-cell transplants, chronic viral infection, organ dysfunction, or advanced age) [98]. It is known that vitiligo can develop in patients with melanoma who have undergone immune stimulation [98]. Since vitiligo is not a common irAE in patients with other cancers, it can be assumed that the development of vitiligo in melanoma patients treated with ICIs may be due to the cross-reactivity of T-cells. However, the cause of irAEs in response to ICIs is still not sufficiently understood. Nowadays, clinical practice guidelines are available that can help to manage with irAEs of different origins and severity [99].

New checkpoint inhibitors are also being tested that could enhance the effectiveness of anticancer therapy; for example, monalizumab, an inhibitory receptor NKG2A (inhibitory receptor NK group 2 member A), which is expressed on NK cells, CTLs, and subgroups of activated CD8^+^ T-lymphocytes [100]. This drug has already shown positive effects in combination with other ICIs [101] and methods of immunotherapy [102].

Unfortunately, not all patients who receive ICIs therapy get the expected results. There are various mechanisms that can cause primary or acquired resistance in patients. These mechanisms can manifest themselves at different stages of tumor recognition and destruction by immune cells [103]. For example, TME can suppress the immune response in various ways. Daniel Chen and Ira Mellman distinguished three basic immune profiles that correlate with immune response to anti-PD-L1/PD-1 therapy, while they examined histological studies of patient tumors [104]. There are three phenotypes by which TME activates immunosuppressive mechanisms: immune desert, immune excluded and immune inflamed. In the first case, there are no T lymphocytes in the TME due to lack of suitable initiation or activation of T-cells. In the second, there are a large number of mediators, chemokines and vascular factors present that inhibit immune cells. Furthermore, in the third, immunosuppression is due to the infiltration of many subtypes of immune cells [104]. Effective strategies directed at overcoming these resistance mechanisms need to be personalized first. Currently, special prognostic markers are actively being developed which could determine the outcome of a therapy before starting it, which can dramatically help in solving this problem by increasing the effectiveness of therapies and reducing the risk of irAEs [105,106]. It is known that immune cells can serve as prognostic markers for assessing responses to ICI therapy. For example, in one study of patients with metastatic melanoma treated with ipilimumab, a significant increase in tumor-infiltrating lymphocytes (TILs) within 3 weeks after starting treatment was associated with positive clinical activity in response to ICI therapy. At the same time, CR was observed in 2.4% of patients, and PR in 17%; PFS was noted in 32.1% of patients [107]. In another study, lymph node and subcutaneous biopsies were taken from patients with metastatic melanomas treated with ipilimumab for analysis of 11 subsets of immune cells residing within the TME. The results of the study showed that CD4^+^ and CD8^+^ T-cells, FOXP3^+^ Treg, CD20^+^ B cells, NKp46^+^ NK cells, lymphocytes expressing markers of activation CD134^+^ and CD137^+^ positively correlated with an increase in OS, the median of which was 30 months [108]. In addition, neoantigens can serve as biomarkers to predict the clinical activity of ICIs. For example, in a retrospective study, an analysis of lung cancer samples was carried out, in which a correlation was found between a high content of neoantigens and a significantly longer OS (*p* = 0.025) [109]. However, it should be noted that in tumor cells, loss of function mutations encoding tumor neoantigens can lead to resistance to ICI therapy [110]. Another prognostic marker for ICIs therapy may be miRNA. One study showed a decrease in PD-L1 expression by breast tumor cells during miR-200 overexpression, which may suggest that these data can be used in the treatment of breast cancer with PD-L1 inhibitors (durvalumab, avelumab, atezolizumab) [111]. Moreover, therapy with ICIs in combination with oncolytic viruses can help overcome tumor resistance by forcing tumor cells to express PD-L1 receptors [112]. However, there are certain problems with developing different prognostic markers, because biopsy samples obtained from the same patient from different tumor sites have shown different levels of biomarkers, due to intratumoral heterogeneity [106].

## 5. Vaccines

Nowadays, vaccines are the most effective treatment against infectious diseases. We have managed to prevent such diseases as smallpox, yellow fever, rubella, polio and measles [113]. There are two types of vaccines: prophylactic and therapeutic. Both types of vaccines aim to elicit specific immune responses. Preventive vaccines act against pathogenic microorganisms or oncogenic viruses (for example, human papillomavirus, HPV) based on attenuated or killed pathogens or virus-like particles (VLP). Therapeutic vaccines are based, for example, on autologous human immune cells or peptides to fight tumor cells.

Several antiviral prophylactic vaccines are already commercially available and are highly effective, including Cervarix^®^, Gardasil^®^ and Gardasil9^®^. Gardasil^®^ was the first vaccine to be approved by the FDA in 2006. It is a vaccine consisting of three recombinant VLPs assembled from the basic capsid protein L1 of HPV types 6, 11, 16 and 18, supplemented with neutral aluminum hydroxyphosphate salt. The second generation vaccine was Gardasil9^®^. Its difference from its predecessor was the wider antigenic spectrum, including HPV types 31, 33, 45, 52 and 58. Cervarix^®^ is composed of VLPs derived from the main capsid protein L1 of the HPV type 16 and 18 formulated in AS04. Cervarix^®^ was approved by the FDA in 2007 [114]. It is not yet known whether immunization with these vaccines will last for a lifetime. It is known that the vaccines Cervarix^®^ and Gardasil^®^ retain their immunogenicity for at least 9 years, and according to some projections, for 20–30 years [115].

Therapeutic vaccines are aimed at activating specific CD8^+^ CTLs. These strategies are based on the interaction of MHC-I epitopes with TAAs. These antigens are delivered in various ways, for example, as adjuvants, to stimulate the presentation of antigen presenting cells (APCs) in vivo. There are several popular therapeutic vaccine strategies. The first strategy is a peptide-based vaccine using screening of amino acid sequences derived from TAAs for potential MHC-I epitopes [116]. The second strategy is to stimulate DCs with TAAs ex vivo, thereby producing an antitumor T-cell response upon presentation of mature optimized APCs [117,118]. The third strategy uses mitotically inactivated tumor cells or their lysate in combination with cytokines such as GM-CSF and/or adjuvants [119]. In addition, it is worth mentioning the experimental vesicle-based vaccines, which also aim to combat tumor diseases [120,121].

### 5.1. Peptide-Based Therapeutic Vaccines

This approach consists of exposing the MHC-I peptide epitopes obtained from TAAs to the cells of the immune system in order to activate CD8^+^ T-cells against their own antigens. Vaccines based on one or more peptides are used either alone or in combination with adjuvants such as Montanide (Seppic’s adjuvant based on three technologies: GEL, IMS and ISA. GEL is a polymer adjuvant based on the dispersion of highly stable sodium polyacrylate gel particles in water; IMS is a combination of micro-emulsions and an immunostimulating compound containing a special fortified mineral oil; ISA is an adjuvant consisting of a mineral oil and a surfactant from the mannide monooleate family, which can be used in the form of various types of emulsions: water-in-oil (W/O), oil-in-water (O/W) or water-in-oil-in-water (W/O/W)), cytokines such as GM-CSF, or peptides loaded into the APC [122,123,124]. A lot of different peptide-based vaccines are used for the treatment of numerous types of cancers such as glioma [125], breast cancer [126], hematopoietic tumors [127], renal cell carcinoma [128] and others. However, the peptide-based vaccination is weakly immunogenic for killing large tumors. In cases where the tumor progresses with metastases, the ORR is about 4% [129].

These vaccines are not suitable for every patient, due to the diversity in human MHC alleles; therefore, peptides with lower affinities for the MHC may be less immunogenic and cannot be presented for the circulating naïve T-cells on ongoing basis. Another limitation lies in rapid degradation of short peptides by serum and tissue peptidases [130]. Specially improving peptide based vaccines for adaptation to a large group of patients like modification of adjuvants, new immunogenic neoantigens and combination with another immunotherapeutic agents may help to avoid these problems [130]. In addition, a vaccine based on short peptides is capable of activating only CD8^+^ T-cells, excluding CD4^+^ T-helper cells, which may limit the functionality of the former. This problem has been overcome using snail lymph hemocyanin (KLH), which is used as an immunogenic xenoantigen to activate CD4^+^ T-cells. Long synthetic peptides including epitopes MHC-I and MHC-II have demonstrated better effectiveness [131]. However, vaccines based on multipeptides or long synthetic peptides, although shown to be more effective, still have a relatively small clinical effect [132]. On the other hand, vaccines based on neoantigens have unique peptide sequences; these are more personalized and hold lower risks for autoimmune generation, which makes them promising targets for activating immune responses [133]. The Dana Farber Cancer Institute (DFCI) recently published promising results in the NeoVax trial in which 4 of 6 patients who were treated with long synthetic neoantigens peptides plus poly IC:LC adjuvant had no recurrence at 25 months following treatment. In addition, two patients were additionally treated with anti-PD-1 inhibitors and experienced complete tumor regression [134].

Perhaps more personalized selection of the antigen and adjuvants, as well as modern delivery methods in combination with other methods of immunotherapy, may help to achieve the greatest effectiveness.

### 5.2. Therapeutic Vaccines Based on Tumor Cells

Vaccines based on tumor cells are often composed of syngeneic, allogeneic or autologous irradiated freshly isolated primary tumor cells, modified, or not, with cytokines such as GM-CSF (GVAX is a vaccine consisting of mitotically inactivated tumor cells genetically modified with the cytokine genome GM-CSF, which has an immunostimulatory effect) or viruses such as the Newcastle disease virus. These vaccines are inoculated subcutaneously in an oil emulsion, such as incomplete Freud’s adjuvant (IFA), or saline solution [135]. These strategies were clinically tested and were in a combinational vaccine in one study in which patients with pancreatic cancer received CRS-207, a recombinant Listeria bacterium expressing TAAs (mesothelin), with or without GVAX, consisting of tumor allogeneic pancreatic cell lines [136,137]. Research results have shown prolonged patient survival with minimal toxicity. However, these combinations did not improve survival in comparison to chemotherapy. In the study, patients were randomly assigned to three groups to receive cyclophosphamide (Cy)/GVAX+CRS-207 (group A), CRS207 (group B), and the physician’s choice of single-agent chemotherapy (group C). The median OS (95% CI) in the primary cohort (N = 213) was 3.7 (2.9–5.3) months in group A, 5.4 (4.2–6.4) in group B and 4.6 (4.2–5.7) months in group C, respectively [138]. Attempts to combine this strategy with other methods of immunotherapy, in particular with ICIs, have also been made. In a similar study, patients with pancreatic cancer received the Cy/GVAX + CRS-207 combination vaccine with or without nivolumab (group B). Median OS in Groups A and B were 5.9 (95% CI, 4.7–8.6) and 6.1 (95% CI, 3.5–7.0) months, respectively. ORR was observed in 4% of group A and 2% of group B [139]. In another study, the use of the GVAX vaccine and ipilimumab in combination with low-dose chemotherapy in phase III clinical trials increased the number of effector CD8^+^ T-cells and increased their lytic function in patients with metastatic prostate cancer [140]. However, no objective responses were obtained with Cy/GVAX + pembrolizumab for patients with progressive mismatch repair proficient (MMRp) colorectal cancer in phase II clinical trials [141].

### 5.3. Therapeutic Vaccines Based on Dendritic Cells

The fact that DCs are one of the main immune cells of innate immunity did not become known immediately, due to their small numbers in circulating peripheral blood. Ralph Steinman and Zanvil Cohn described the dendritic cells of lymphoid organs in 1973 [142], and over the next few years, evidence has accumulated that has led to DCs being regarded as the main APCs. DCs are derived from hematopoietic stem cells (HSCs), both myeloid and lymphoid progenitors. Due to the high expression of MHC-I and MHC-II, as well as co-stimulatory molecules, especially CD80 and CD86, these cells are superior to macrophages in their effectiveness of “professional” antigen presentation. Moreover, only DCs are capable of activating naive T-cells [143,144], which prompted scientists around the world to search for an effective vaccine for the treatment of cancer. For this purpose, it was necessary to get a better understanding of the histogenesis of dendritic cells, the regulation of the innate and adaptive immune system, the tumor microenvironment, and also to use the necessary genetic methods to achieve maximum efficiency and increase the antitumor immunotherapeutic potential.

There are several ways to present targeted information for DCs (ex vivo). The most common way is to use peptides, proteins, tumor cells or their lysates [145,146]. Tumor cells need to inactivate by 24-h TNF sensitization, gamma irradiation and 48-h cultivation in serum-free medium. After that DCs cultivate with inactivate tumor cells for a few hours [147]. Tumor lysate is obtained by four freeze (−80 °C) and thaw (room temperature) cycles. Then DCs cultivate with tumor lysate overnight [148]. Peptides are synthesized by automated solid phase synthesis and cultivate with mature DCs for 4–6 h [149]. Alternative methods are viral vectors and transfection of DNA and mRNA encoding TAAs for presentation and induced maturation of DCs by lentiviral transduction and electroporation, respectively [150,151,152]. In phase IB clinical trial, DCs electroporated with mRNA encoding proteins for maturation and with mRNA encoding fusion proteins of a HLA-class II targeting signal and a melanoma-associated antigen [153]. It should be noted that there is a method to bypass the long and costly generation of activated DCs ex vivo by selectively delivering antigens to DCs in vivo using chimeric proteins consisting of mAbs, specific for certain DCs surface molecules, and antigens using appropriate adjuvants [154]. However, for the development of such vaccines, many factors must be taken into account, such as the choice of the appropriate antigen, the required receptor on the surface of the DCs, and the choice of an effective adjuvant.

Researchers at Rockefeller University, conducting clinical trials with DC-based vaccines using apoptotic tumor cells of the prostate (LNCaP/PC3), noted an interesting feature in preparing DCs before vaccination. Two different methods were used to isolate DCs precursors: adhesion of peripheral blood mononuclear cells (PBMCs) to plastic (DCs adhesion method) and CD14^+^ cell selection using CliniMACS antibody-conjugated granules (DCs selection method). They found that patients who received the vaccine from DCs by adhesion showed increased T-cell proliferation in response to the vaccination, compared to the vaccine from DCs obtained by the selection method. The scientists concluded that the presence of activated lymphocytes may enhance the response seen in patients receiving the vaccine using an adhesive preparation method. This is an important finding considering the variety of methods used in preparing vaccine studies and different responses after vaccination [155].

It is also worth mentioning that there are several different routes for administering DCs-based vaccines: intranodal, intradermal, subcutaneous, intravenous, and intralymphatic administration. This is certainly an important aspect, because the effect can vary depending on the injection site, and this cannot be ignored. For example, in one study, it was shown that following intradermal administration, the level of migration of the injected DCs to the lymph nodes was very low [156].

In the first phase of clinical trials of DCs-based vaccines, the researchers noted the full feasibility of this therapy and its safety, due to the lack of toxicity compared to other methods of treating malignant neoplasms. Thus, according to one of the first reported studies, patients with low grade follicular B cell lymphoma resistant to chemotherapy were immunized with DCs loaded with target clonal immunoglobulin antigens, followed by injections of this idiotypic protein and KLH [157]. All patients had cell proliferative reactions, one patient had partial tumor regression, another had complete regression, and the third had no signs of the disease. In another study, 16 melanoma patients received a dendritic vaccine loaded with a cocktail of gp100 peptides, MART-1 (melan-A or melanoma antigen recognized by T-cells 1), tyrosinase, MAGE-1 (melanoma-associated antigen 1), or MAGE-3 (melanoma-associated antigen 3) and 4 patients were treated with a dendritic vaccine loaded with tumor lysate. In both cases, KLH was added with antigen loading. Tumor regression was observed in 5 out of 16 patients, among them 2 patients showed complete regression lasting 15 months [158]. However, most of those trials were unique, differing from each other in various ways, ranging from the way the vaccine was administered and the different numbers of cells injected, to the different antigen loading strategies, which makes direct comparisons difficult.

In subsequent clinical trials, the vaccine had a certain biological activity, but its therapeutic efficacy left much to be desired [159,160]. In developing an effective therapeutic vaccine, researchers faced certain difficulties, since cancer vaccines target antigens associated with a tumor, such as differentiation antigens or overexpressed tumor antigens. However, negative selection in the thymus limits the generation of cytotoxic T-lymphocytes with high avidity directed against these antigens [161]. In order to solve this problem, researchers have resorted to various options. In one study, scientists at the National Cancer Institute (NIC) identified neoantigens recognized by TILs that mediate regression in patients with metastatic cancer (NCT03300843). In another study, scientists use certain adjuvants, namely curdlan sulfate (CS), for an immunostimulatory effect and activation of dendritic cells [162].

There is currently an FDA-approved therapeutic vaccine, Sipuleucel-T, for the treatment of metastatic prostate cancer. This vaccine is based on DCs loaded with a recombinant fusion protein (prostatic acid phosphatase (PAP) and GM-CSF fusion). Sipuleucel-T demonstrated a median improvement in OS of 4.1 months compared with placebo in men with metastatic castration-resistant prostate cancer (mCRPC) [163]. Sipuleucel-T is currently being tested in a phase II clinical trial versus ipilimumab in men with chemotherapy naive mCRPC (NCT01804465), in combination with radium-223 in men with asymptomatic or minimally symptomatic mCRPC (NCT02463799), and in combination with glycolyzed IL-7 in men with mCRPC (NCT01881867).

The development of a therapeutic vaccine based on DCs is an important milestone in the history of immunotherapy. Many different clinical trials are underway around the world in which DCs-based vaccines show their safety and efficacy in combination with other immunotherapy agents (NCT01302496, NCT01876212, NCT03152565, NCT00266110). Moreover, these vaccines are appropriate for patients with less weakened immune systems so that a higher immune response can be expected. Patients without signs of cancer progression can better respond to immunotherapy, rather than patients who no longer respond to standard therapy. A personalized approach is also needed to select the correct loading method for DCs and the necessary adjuvant for an immunomodulatory effect. Considering moderate therapeutic effects, this vaccine can become a kind of “reference” for comparative analysis for new methods of immunotherapy.

### 5.4. Therapeutic Vaccines Based on Extracellular Vesicles

In the past few years, the global scientific community has begun to actively discuss the therapeutic potential of extracellular vesicles (EVs). The existence of EVs was discovered in 1983–1985 [164,165]. Since then, a large amount of data has appeared on their presence in healthy people and people with various diseases, as well as on their composition and properties. EVs are membrane-bound structures of various sizes and are secreted by different types of cells. They contain a wide range of bioactive molecules, including proteins, lipids, nuclear and mitochondrial components. EVs perform a number of important functions related to cell-to-cell communication, influencing multiple cellular processes. This process is mediated by the ability of EVs to fuse with recipient cells through endocytosis. It is due to the properties described above that EVs are considered as a potential immunomodulatory agent, as well as a promising vector for the delivery of various types of antitumor agents [120]. Several anti-tumor strategies have been developed using EVs. These involve the release of EVs from immune cells, mesenchymal stem cells (MSCs) or tumor cells, loading them with anticancer agents and the direct effect of EVs on tumor cells. Another strategy is culturing EVs derived from tumor cells with autologous immune cells for immunomodulatory effects and subsequent administration to a patient.

EVs derived from MSCs are now increasingly viewed as a potential therapeutic tool in regenerative medicine, for example in the prevention of autoimmune diseases. One study demonstrated that EVs derived from MSCs inhibit APC activation and suppress Th1 and Th17 development [166]. In addition, in a different study, EVs derived from umbilical cord mesenchymal stem cells (hUC-MSCs) increased the Treg ratio, while decreasing the number of Th17 [167].

EVs derived from immune cells may have real therapeutic effects in the treatment of cancer. For example, it was shown that EVs derived from NK cells (NKEVs) are able to exert a stimulating effect on PBMCs, as well as increase the fraction of CD56^+^ NK cells, which suggests a potential support for cancer therapy [168]. However, clinical trials conducted with such EVs indicated that native EVs are not sufficient to exert the desired effect in vivo; but, considering their safety when administered, as well as their ability to modify and target a specific cell, EVs make an interesting tool for the treatment of cancer [120], neurodegenerative [169], inflammatory and autoimmune diseases [170].

A significant amount of evidence has emerged that tumor cell-derived EVs have an immunosuppressive effect on the immune system rather than an immunomodulatory one. Culturing EVs derived from tumor cells with PBMCs revealed that EVs shift the differentiation of monocytes into DCs towards the development of myeloid cells, which have a suppressive effect on T-cells [171]. In another study, it was shown that EVs derived from glioblastoma can modify cells of monocytic origin, which acquire features similar to tumor phenotypes observed in patients [172]. Numerous studies have led to the conclusion that tumor-derived exosomes are capable of modulating the cellular biology of myeloid suppressor cells (MDSCs). Furthermore, these MDSCs promote tumor progression through the production of suppressive molecules [173]. Potential applications of tumor EVs include the diagnosis of cancer through proteomic and phosphoproteomic biomarkers, as well as use of EVs in targeted drug delivery. In this way, such an EVs-based delivery system demonstrates low toxicity, stability, and increased biocompatibility [174].

## 6. Oncolytic Viruses

There are a lot of recombinant viral vectors based on adenovirus, herpes simplex virus, smallpox virus, Coxsackie virus, maraba virus, poliovirus measles, and Newcastle disease virus. All these viruses have been used in various strategies for the treatment of cancer [175]. Most viruses are composed of three elements: the genome, the capsid, and the lipid envelope. Each type of virus has an individual structure, which is reflected in its variability when used as a tool against a tumor. For example, large eukaryotic transgenes can be encoded in DNA viruses to provide improved therapeutic activity or immune modulation, or DNA polymerases for more efficient replication. RNA viruses, for example, reoviruses, having a less capacious genome can, however, penetrate the blood–brain barrier, targeting a tumor in the central nervous system [176]. Research in the field of oncolytic viruses (OVs) as a method of immunotherapy has developed since the early 1950s [177], but over the last 15 years work has expanded more rapidly into the oncolytic capabilities of this therapy. OVs are a versatile tool for treating malignant diseases. Their antitumor activities have a wide arsenal of possibilities for the natural interaction of viruses and the immune system with tumor cells. OVs trigger selective replication in tumor cells and induce their subsequent death, thereby spreading TAAs and other factors, activating the innate and adaptive immune system [178]. Some wild-type OVs are able to recognize highly expressed receptors on the surface of tumor cells or other abnormal products or pathways in tumor cells, thereby targeting them. For example, coxsackievirus A21 (CVA21) has a natural tropism for tumor cells, recognizing highly expressed receptors and intercellular adhesion-1 molecules (ICAM-1/CD54) on the surface of tumor cells for subsequent penetration, replication, and their elimination [179]. A CVA21-based OV (CAVATAK^®^) has been shown to be safe and able to stimulate an anti-tumor immune response in the treatment of melanoma patients (NCT01227551, NCT01636882). Another wild-type virus, Parvovirus H1 (H1-PV), exhibits tumor selectivity for replicative and transcriptive factors and a defective type I IFN-mediated antiviral pathway in tumor cells [180]. ParvOryx (wild type H1-PV) has shown safety and evidence of immunogenic activity in the treatment of glioblastoma in phase I/IIa clinical trials. The study involved 18 patients and the median PFS was 15.9 weeks, and the median OS was 464 days [181]. In addition, the wild-type reovirus exhibited tropism towards tumor cells due to their abnormally activated Ras pathway [182]. Reolysin^®^ (pelareorep) is a patented variant of reovirus that has received FDA approval as an orphan drug for the treatment of gastric and pancreatic cancer [183].

However, this strategy has a problem with the antiviral immune response. The immune response is quite focused on viral antigens, which reduces the effectiveness of therapy. Many options and strategies have been tried in order to solve this problem. For example, DCs, MSCs, T-cells, and cytokine-induced killers (CIKs) are used as host cells into which the virus is loaded. These host cells not only protect the virus from immune neutralization, but can also reduce the uptake of the virus by the reticuloendothelial system and bring the virus to a potential tumor site [184]. Furthermore, the replicative ability of viruses in tumor cells is a very important part of the therapy. Some viruses have an innate tropism for tumors, but other viruses require genetic editing to selectively infect tumor cells. Various strategies are used to increase the selectivity of viruses and efficiency of replication [178]. Genetic modifications of OVs are mainly reduced to the removal of virulence genes to ensure safety, as well as the inclusion of foreign genes to increase the antitumor effect and tumor selectivity [185].

Various therapeutic or immunomodulatory genes are used to enhance antitumor activity. Recombinant adenovirus with p53 gene insert (Oncorine) was the first approved drug. This OV was initially developed in China in 1999, and it finally received a Good Manufacturing Practice (GMP) license for the treatment of head and neck cancer in 2006 [186]. In a Phase III clinical trial, patients with head and neck squamous cell carcinoma (HNSCC) were randomized to an experimental group (Oncorine + chemotherapy) and a control group (chemotherapy). CR was obtained in 11.5% of patients in the experimental group and 3.7% of patients in the control group. PR was obtained in 67% and 35% of patients, respectively. These data suggest a higher response rate in patients receiving chemotherapy in combination with Oncorine [187]. A popular strategy for enhancing anti-tumor activity is the insertion of cytokine or chemokine genes expressed by the virus. This approach using GM-CSF was approved by the FDA in 2015. Talimogene laherparepvec (T-Vec), an attenuated herpesvirus encoding GM-CSF, increased median survival by 5 months in patients with metastatic melanoma [188]. OVs are also able to suppress the angiogenic abilities of tumors. For this, the endostatin gene is used, as well as genes encoding the VEGI and VEGFR-1 proteins [189,190,191].

A large number of preclinical and clinical trials of this therapy are underway in combination with traditional cancer treatments such as radiotherapy and chemotherapy, and research studies are underway with new immunotherapy methods such as ICIs or CAR T-cell therapy [178,192,193]. Reolysin^®^, in combination with carboplatin and paclitaxel in the treatment of progressive malignant melanoma, has shown improved antitumor efficacy. ORR was 21%, and median PFS and OS were 5.2 and 10.9 months, respectively [194]. ONYX-015 (E1B-deleted adenovirus, replicates in p53-deficient human tumor cells) in combination with cisplatin and 5-fluorouracil increases OR up to 65% compared to monotherapy, where OR reached 15% [195]. In a phase I/II clinical trial for the treatment of squamous cell carcinoma of the neck and head, patients were injected intravenously with T-vec along with cisplatin chemoradiotherapy, followed by neck dissection in 6–10 weeks after treatment. OR was observed in 82.3%, while complete remission was confirmed in 93% of patients [196]. OVs combine with ICIs to create a synergistic effect. OVs can attract CD8^+^ T-cells and NK cells to TME and induce PD-L1 on the surface of tumor cells, thus eliminating the lack of immune cells in the TME and facilitating blocking for ICIs, thereby enhancing the antitumor effect [192]. This combination has already demonstrated itself in phase Ib clinical trials in the treatment of patients with progressive melanoma [197]. This study evaluated the effect of intratumoral T-vec injection on cytotoxic T-lymphocyte infiltration and the therapeutic efficacy of pembrolizumab. The ORR was 62%, while the CR was 32%. Patients who responded to therapy showed an increase in CD8^+^ T-cells, as well as increased expression of the PD-L1 protein and expression of the IFN-γ gene on some cellular subgroups of tumor cells. The combination of OVs with CAR T-cell therapy may be the key to overcoming multiple intratumoral barriers. OV-induced lysis of tumor cells promotes tumor infiltration for CAR T-cells and also releases TAAs, thereby generating a T-cell response to these antigens, providing synergy between CAR T-cells and cytotoxic T-lymphocytes in the elimination of heterogeneous tumor cells. The combination of OVs with CAR T-cell therapy is currently being tested in mice, showing a definite antitumor effect [193].

## 7. CAR T-Cell Therapy

CAR T-cell therapy is one of the most successful experimental and rapidly developing innovative approaches to the treatment of malignant neoplasms, with a large number of registered clinical trials. The first scientific works were mostly fundamental, and scholars studied the mechanisms and role of the T-cell receptor (TCR), MHC and other coreceptors [198]. The T-lymphocyte carries a TCR, which consists of two polypeptide chains that belong to the immunoglobulin superfamily, but they are organized more simply (lack of an Fc fragment). Antibodies recognize antigenic epitopes of native protein molecules on the cell membrane. At the same time, TCR recognizes peptide fragments of antigens in complex with the MHC molecule on the cell surface. The TCR dimer chains are linked to the polypeptide chains of the CD3 complex. An activating signal formed by the interaction of TCR and peptide-MHC is transmitted through the ζ-chains of the CD3 complex for an intracellular signal. In order for TCR to recognize antigens without the participation of MHC and react on them, therefore, it was proposed that combining the antigen-binding variable domain of antibodies with the constant domain of TCR in one polypeptide may be beneficial, due to the similarity in the structure and organization of TCR and immunoglobulin [199]. The result of such experiments was first generation CAR.

CAR is a recombinant receptor located on the surface of a T-cell, which determines the specificity of this cell for the corresponding antigen and activation of the T-lymphocyte. A first generation CAR is an antigen binding variable part of a mAb (scFv) (extracellular part) and a native TCR containing a ζ-chain fragment of the CD3 complex (intracellular part). The efficacy of the first experiments with the participation of such structures was low [200,201,202,203]. Therapeutic cells underwent apoptosis after several divisions. Later it became known that for the full activation and proliferation of T-lymphocytes, it is necessary to have an interaction between the antigen and TCR in combination with MHC, as well as the interaction of costimulatory receptors (CD28, 4-1BB, OX-40) on the surface of T-cells with the corresponding ligands (CD80/86, 4-1BBL, OX-40L) on the APC. Thus, the second generation of CAR appeared, containing one costimulatory domain in addition to the scFv and CD3z domain.

Treatment efficacy of CAR varied. This therapy was most effective with the use of CD19 T-cells in B-linear ALL, and it also performed well in B-cell lymphoma and CLL. Mainly second generation CAR T-cells were used, because first generation CAR was not as effective, but the use of the costimulatory domain varied from study to study. A University of Pennsylvania study used CD19 T-cells with TCR and 4-1BB co-stimulatory domains in patients with chemotherapy-resistant or refractory ALL. In a cohort of 30 people, a single dose of autologous T-cells (1–5 × 10^8^ CD19 CAR T-cells) were administered by split dosing: 10% on day 1 (1–5 × 10^7^), 30% on day 2 (3 × 10^7–^1.5 × 10^8^) and 60% on day 3 (6 × 10^7^–3 × 10^8^), and the following results were obtained: patients with CR—26.7%; patients with CR with incomplete recovery of blood parameters—33.3%, with a median follow-up of 6 months [204]. In another study, therapy was performed in patients with multiple recurrent or refractory CLL. CD19 CAR T-cells (CTL019) with a 4-1BBz costimulatory domain were administered to patients with recurrent/refractory CLL at doses ranging from 0, 14 × 10^8^ to 11 × 10^8^ CTL019 cells (median, 1,6 × 10^8^ cells). The cohort of patients who received CAR T-cell therapy consisted of 14 patients. Prior conditioning included cyclophosphamide and fludarabine, or pentostatin and cyclophosphamide, or bendamustine. ORR for treatment in patients with CLL was 57%, of which CR 50% and PR 50%. At the same time, CAR T-cells were preserved and remained functional for 4 years in the first two patients who achieved CR [205].

In the second generation CAR, two co-stimulatory domains are most commonly used: CD28 and 4-1BB. Some studies have compared them with each other. In one study, scientists showed that CD28 more effectively resulted in tumor elimination with fewer cells [206]. In another study, longer persistence of 4-1BB CAR T-cells was demonstrated [207]. This factor may play a significant role in favorable clinical outcomes in cancer treatment [208]. However, no direct in vivo comparisons have been made to validate this.

To create a third generation CAR, two costimulatory domains were used in the CAR construct. Both costimulatory domains showed increased activity and signaling of CAR T-cells. They mediate a complete response in patients with progressive CD19-positive hematologic malignancies. Third generation CAR incorporating two co-stimulatory domains in their constructs, CD28 and 4-1BB, have demonstrated superior proliferative ability, and more robust survival and antitumor responses in vitro and in vivo in comparison to second generation CAR containing CD28 or 4-1BB [209,210,211]. The first phase I/IIa clinical study using third generation CAR T-cells enrolled patients with leukemia and lymphoma who received severe prior therapy with significant comorbidities [212]. In general, the use of the third generation CAR was safe. Overall, 40% of patients had an initial CR (of which: 60% lymphoma, 40% ALL), 20% of patients with lymphoma were in remission after 3 months and 13% of patients were still alive at the time of publication. The authors of this study concluded that third generation CAR may be effective in patients with B-cell lymphoma with only mild toxicity. Another study evaluated the efficacy of third generation CAR by simultaneously administering second generation CAR T-cells (CD28 only) and third generation CAR T-cells in 16 patients with recurrent or refractory non-Hodgkin’s lymphoma [213]. Compared to cells transduced with the second generation CAR vector, the third generation CAR T-cells showed superior proliferation and longer persistence, especially in patients with low disease severity and low circulating B cells. Cytokine release syndrome (CRS) was observed in 37% of patients, but it was mild, and no patient required anti-IL-6 therapy.

The University Hospital Heidelberg is the first institution in Germany to conduct a Phase I–II third generation CAR T-cell test. This study is being conducted for the treatment of adult and pediatric patients with refractory or recurrent ALL, CLL, as well as diffuse large B-cell lymphoma, follicular lymphoma, and mantle cell lymphoma with autologous T-lymphocytes transduced with a third generation CAR retroviral vector, aimed at CD19 [214]. The main goal of this study is to assess the safety and feasibility of increasing doses of CD19 CAR T-cells (1–20 × 10^6^ cells/m^2^) after lymphodepletion with fludarabine and cyclophosphamide. Patients will be screened for the presence of CRS and neurotoxicity, i.e., immune effector cell-associated neurotoxicity syndrome (ICANS). It will also assess the functional antitumor activity and survival of CD19 CAR T-cells in vivo (NCT03676504). However, the number of patients treated with third generation CAR T-cell therapy is insufficient to fully and conclusively prove its efficacy, and further studies are needed to thoroughly evaluate CD19-targeted third generation CAR T-cells.

The fourth generation CAR T-cells are based on the second generation construct with the inclusion of a cassette for the inducible expression of a transgenic cytokine, in this case IL-12. It is also known as T-cell redirected for universal cytokine-mediated killing (TRUCKs) [215]. It is assumed that this design will overcome the limitations associated with targeting solid tumors with a heterogeneous phenotype. CAR-mediated T-cell activation and inducible release of IL-12 in the tumor will enhance T-cell activation and stimulate the innate immune system to destroy antigen-negative tumor cells [216]. However, in the first clinical trials, this construct caused toxic effects, including liver dysfunction, high fevers and sporadic life-threatening hemodynamic instability, which may be mediated by high levels of IL-12 secretion, and this construct did not provide the desired therapeutic effects [217].

There is also another fourth generation construct (4SCAR) that includes a secretory signal peptide, scFv (e.g., CD19) and transmembrane domains CD28, CD27 and CD3z, 2A sequence, and an inducible switch caspase 9 (iCasp9) [218]. iCasp9 was designed to improve the safety and reduce toxic effects of T-cells in adoptive therapy [219]. In one study of a 67-year-old man with primary central nervous system lymphoma (PCNSL), after a second relapse, it was decided to proceed with chemotherapy with lymphodepletion followed by infusion of fourth generation CD19-CAR and CD70-CAR T-cells (4SCART19 and 4SCART70). Neither CRS nor ICANS arose during treatment and follow-up, and at the time of publication, the patient had a disease-free survival rate of over 17 months [218]. Comparative characteristics of CARs of different generations are shown in Table 2.

The first FDA approved drug involving CAR T-cells was tisagenlecleucel (Kymriah) in ALL patients up to 25 years of age in 2017 [220]. Kymriah helped to reach complete remission in ~60–90% of patients with refractory or relapsed ALL. However, there is still a high risk of developing CRS and ICANS. Fever, rigors, tachycardia, hypotension, tachypnea, hypoxemia, and other signs of systemic inflammation are commonly observed in CRS. Such CRS are reversible in most patients, but there are also grade ≥3 or 4 CRS, sometimes with fatal cases, that have been reported. Severe grade ≥3 CRS was characterized by profound hemodynamic instability, capillary leak syndrome, and consumptive coagulopathy. The development of ICANS is more complicated. Neurological symptoms such as headache, tremor, speech impairment, delirium, confusion, impaired consciousness have been observed [221]. Endothelial damage, microglial activation, parenchymal necrosis, multifocal microhemorrhages and CAR T-cell infiltration of the brain in patients with fatal cerebral edema after CD19 CAR T cell therapy have been reported. However, severe ICANS is rarely observed in the absence of severe CRS [222]. It is important to thoroughly count the CAR T-cell dose to prevent severe CRS and ICANS. Moreover, administration of corticosteroids, tocilizumab (IL-6 receptor-directed antibody) and anakinra (IL-1 receptor antagonist) may reduce CRS and ICANS severity [221].

Several promising strategies have been developed to prevent CRS and ICANS in patients using CAR T-cell therapy. The most popular strategies are based on the suicide gene switch, synthetic notch receptors and bispecific T-cell engager [223]. Suicidal genes include herpes simplex virus thymidine kinase (HSV-tk) and iCasp9. HSV-tk is used in combination with ganciclovir (GCV), namely, HSV-tk phosphorylates GCV, resulting in the formation of GCV triphosphate, which inhibits DNA synthesis, leading to the death of CAR T-cells. This technique is most commonly used to infuse donor lymphocytes after hematopoietic stem cell transplantation to prevent graft-versus-host disease (GVHD) [224]. However, HSV-tk is also used in CAR T-cells, for example, in acute myeloid leukemia or multiple myeloma [225]. Although this method has a number of disadvantages associated with the immunogenicity of the virus [226] and its slow activation [227].

iCasp9 contains a modified human caspase 9 fused to the human binding protein FK506 (FKBP), which dimerizes when a chemical dimerization inducer (CID) (AP1903) is introduced, thereby triggering a signaling cascade that activates cell apoptosis [228]. This strategy performed well in preclinical trials, eliminating second generation CAR T-cells within 3 days [229], and also demonstrated a 90% reduction in third generation CAR T-cells 12 h after CID administration for two days [230]. Clinical trials using iCasp9 against solid tumors are currently underway (NCT01822652, NCT01953900, NCT02107963) [228].

A new class of synthetic Notch receptors (synNotch) helps to exhibit selective cytotoxicity to tumor cells, avoiding damage to normal tissues. This selectivity is achieved by targeting CAR T-cells to two different antigens. The synNotch receptor first recognizes one tumor antigen, resulting in the release of a transcriptional activator domain in the T-cell, followed by expression of a CAR targeting another tumor antigen. Thus, CAR T-cells exhibited selective cytotoxicity to ROR1^+^ tumor cells, while avoiding ROR1^+^ stromal cells [231]. Another strategy is that CAR T-cells do not directly recognize antigens on target cells, but bind to fluorescein isothiocyanate (FITC) molecules, thereby exhibiting cytotoxicity to cells via bispecific molecules, which limits their cytotoxicity towards normal tissues [232]. For example, trastuzumab conjugated with the FITC molecule forms a bispecific antibody (Ab-FITC) that redirects anti-FITC CAR T-cells to HER2^+^ tumor cells, inducing antitumor activity in vitro and in vivo [233].

Along with the problem of CRS and ICANS, there are also difficulties with relapse after effective treatment with CD19 CAR T-cell therapy. Basically, it can manifest itself with a decrease in the function of T-cells or their disappearance, or with a loss of expression of the surface antigen; in this case, CD19. In order to overcome the loss of the target antigen, the CAR molecule was engineered to recognize multiple antigens by linking two binders on a single molecule (tandem CAR) [234]. Such bispecific CARs as CD19-CD20 or CD19-CD22 can increase the efficacy of the CAR T-cell therapy [235,236]. In a phase one trial, patients with resistance to CD19-targeted immunotherapy received CD19-CD22 CAR T-cells. Complete remission was obtained in 73% of patients and the median remission duration was 6 months [236]. The crucial role of antigen density in the regulation of receptor function was also assessed, since relapses were associated with a decrease in the density of CD22 sites. Another approach to overcome the loss of the antigen was invented by scientists from the United States and China who developed second generation CAR T-cells against another marker of B-specific cells, namely the B-cell activating factor receptor (BAFF-R), a target for cancer immunotherapy, which is not yet fully implemented [237]. The experiments were carried out in vitro and in vivo on xenographic mouse models. 

The efficacy of CAR technology for solid tumors is much lower compared to CD19 CAR T-cell therapy. Therapy in solid tumors is difficult due to several factors, such as the vessel wall and stroma, which make it difficult for the modified T-cells to contact the target cancer cells, as well as the immunosuppressive microenvironment that the tumor creates. However, research in this area does not stop and solutions for overcoming these problems are found in different ways; for example, by combining CAR T-cells with chemotherapeutic agents that modulate the antitumor response and, at the same time, reduce immunosuppressive factors [238]. Another strategy is to use VHH-based CAR constructs (instead of scFv), whose binding domains are derived from camel single-domain antibodies, for anticancer therapy [239]. Heavy chain antibodies (hcAb, two polypeptide chains, 75 kDa) of natural origin, containing a highly stable and soluble single antigen-binding V-domain, designated VHH or nanobody (15 kDa), have opened new horizons in anticancer therapy. Compared to conventional antibodies, nanobodies are smaller, more soluble and more stable, and can easily penetrate dense tissues such as solid tumors. One study has shown the efficacy of VHH-based CARs as in vitro anti-angiogenic therapy. The authors used second-generation nanbody-based CAR T-cells (VHH) targeting VEGFR2-expressing tumor cells [240].

To overcome the mechanisms of immunosuppression and in order to enhance efficacy, scientists from the University of Erlangen-Nuremberg have generated T-cells expressing two additional receptors (TETAR) that are capable of secreting cytokines and exhibiting cytotoxicity [241]. However, data on clinical trials using these technologies are not yet available. One study used a third generation CAR with an activating NKG2D receptor against human colorectal tumor cells (LS174T and HCT-116). The NKG2D CAR consists of the CD8α signal sequence, the extracellular part of the human NKG2D receptor, the CD8α hinge region, the CD28 transmembrane and intracellular domain, the human 4-1BB intracellular signaling domains, and CD3ζ. NKG2D CAR T-cells have demonstrated specific and effective cytotoxicity against human colorectal cancer cells both in vitro and in vivo in xenographic mice models [242]. One of the obvious ways to increase therapeutic efficacy is the combination of CAR T-cell therapy and PD-L1 inhibitors [243].

## 8. Conclusions

Of all the above methods, it is necessary to select the most optimal approaches, taking into account the positive and negative aspects of each method. Cytokine therapy has had a great effect on the functionality of immune cells, which can contribute to the active elimination of malignant neoplasms. On the other hand, this advantage is also a disadvantage of this method due to the surge in autoimmune activity and, as a consequence, the appearance of serious side effects. The discovery of the antitumor potential of monoclonal antibodies has given new hope for successfully fighting against cancer. They have demonstrated their therapeutic potentials, but the risk of serious side effects is still high and the emergence of tumor resistance to this technique does not allow for a positive result. In addition, various variants of therapeutic vaccines have shown their antitumor capabilities and safety in use, but their therapeutic effects are still low. The ability of oncolytic viruses to destroy tumor cells selectively, thereby stimulating immune cells, reveals their antitumor potential; however, despite different ways of overcoming antiviral immunity, their therapeutic effect is also not high. With the help of CAR T-cell therapy, mankind has made a huge step in the treatment of malignant hematopoietic and lymphoid diseases. However, this method of immunotherapy cannot cope with solid tumors, due to their heterogeneous phenotypes and immunosuppressive microenvironment. In conclusion, it should be noted that a combination of immunotherapy methods with traditional therapies and new targeted methods, used together, may be able to give an effective result and overcome the limitations of any particular approach. Combinations such as CAR T-cell therapy with ICIs or cytokine therapies, or oncolytic viruses in combination with ICIs that will be used in a personalized manner, may demonstrate the desired effect. Therefore, in subsequent clinical trials, it is extremely important to learn how to determine individually which combination of methods will give the best results in the treatment of a particular oncological disease for each patient.

## Figures and Tables

**Figure 1 biomedicines-08-00621-f001:**
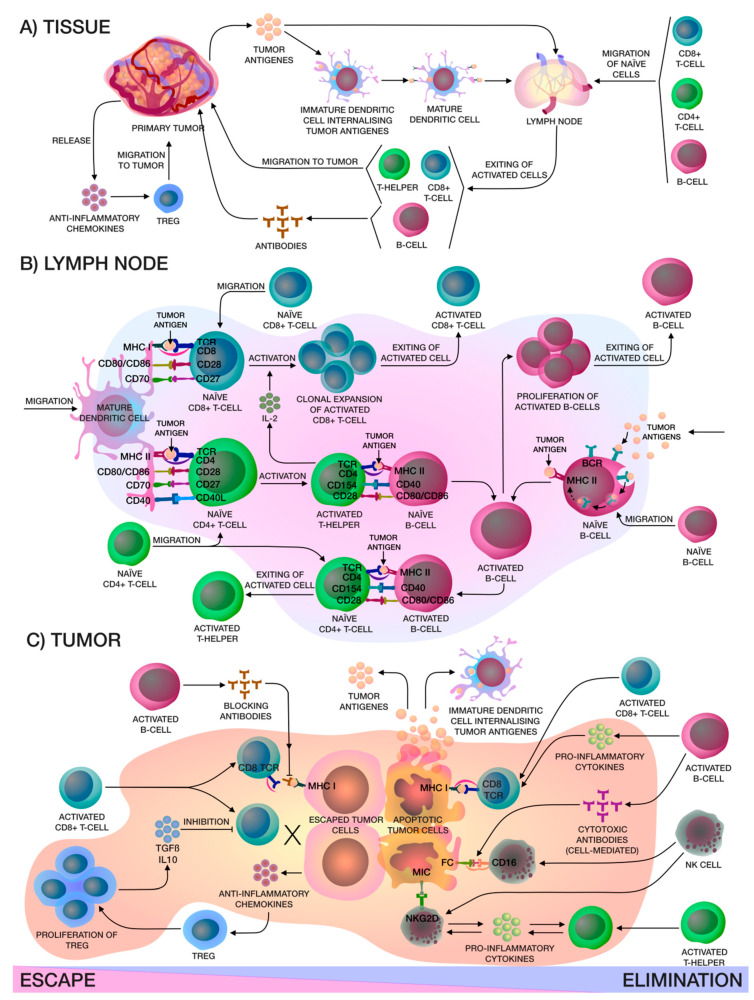
(**A**) Localized in the adjacent tumor tissue, the immature CD14^+^ dendritic cells (DCs) internalize the captured tumor antigen, become mature CD14^+^ DCs and migrate to the lymph node for presentation to naïve lymphocytes, in particular CD8^+^ T-cells, CD4^+^ T-cells and CD19^+^ B-cells, which are in constant recirculation. After that, activated lymphocytes exit from the lymph node, activated CD19^+^ B-cells produce corresponding antibodies and activated CD4^+^ T-helpers and CD8^+^ cytotoxic T-lymphocytes (CTLs) migrate to the tumor niche. At the same time, the primary tumor releases chemokines that attract FOXP3^hi^ Tregs. (**B**) In the lymph node, mature CD14^+^ DCs present tumor antigens to naïve lymphocytes, activated CD4^+^ T-helpers release interleukin (IL)-2 for clonal expansion of activated CD8^+^ CTLs and help naïve B-cells on the border of the B-cell zone of the lymph node. In addition, naïve B-cells may capture tumor antigens from the lymph and, after internalizing, activated B-cells proliferate and, furthermore, they can present tumor antigens to naïve CD4^+^ T-cells. All in all, activated lymphocytes leave the lymph node. (**C**) There are two mechanisms that occur in the tumor niche: escape or elimination. Elimination of the tumor, in the main, occurs due to activated CD8^+^ CTLs and CD56^+^ natural killer (NK)-cells. Activated CD19^+^ B-cells only help activated CD8^+^ CTLs and CD56^+^ NK-cells by releasing interferon-γ (IFN-γ) and IL-12, which enhance the cytotoxic effects, and also produce cytotoxic antibodies that cause a cell-mediated immune response. Activated CD4^+^ T-helpers and CD56^+^ NK cells release pro-inflammatory cytokines like IFN-γ and tumor necrosis factor alpha-α (TNF-α), that enhance each other. Apoptotic tumor cells release antigens that stimulate immune cells. On the other hand, primary tumors can inhibit effector immune cells by releasing soluble antigens and anti-inflammatory chemokines that attract FOXP3^hi^ Tregs, thereby creating a tumor microenvironment. Moreover, antibodies produced by activated CD19^+^ B-cells may block access to tumor cell for activated CD8^+^ CTLs.

**Figure 2 biomedicines-08-00621-f002:**
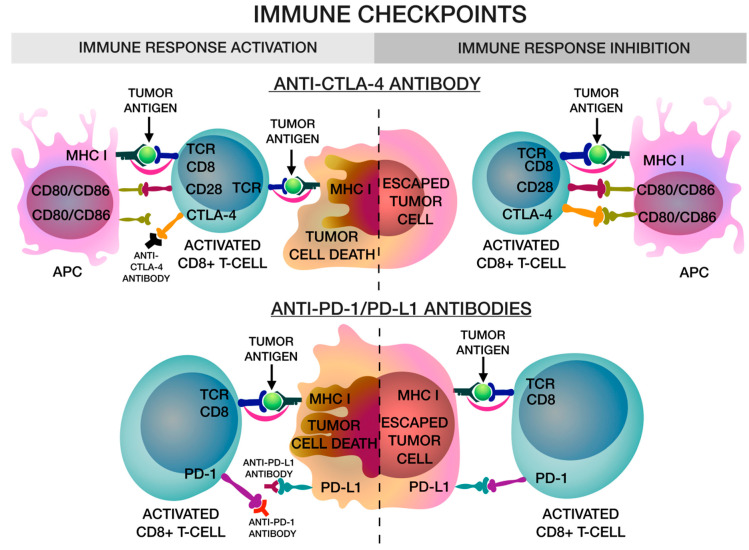
Binding of CD80/CD86 and cytotoxic T-lymphocyte-associated protein 4 (CTLA-4) keeps CD8^+^ T-cell in an inactivate state, that do not let them kill tumor cells. An anti-CTLA-4 antibody blocks the binding and activated CD8^+^ CTLs can eradicate tumor cells. The binding of programmed cell death 1 (PD-1) and programmed cell death ligand 1 (PD-L1) helps tumor cells avoid apoptosis, inhibiting activated CD8^+^ CTLs. Anti-PD-1 and anti-PD-L1 antibodies block the binding and enable activated CD8^+^ CTLs to kill tumor cells.

**Table 1 biomedicines-08-00621-t001:** Immune checkpoint inhibitors for cancer therapy.

Inhibitor Type	Drug	Cancer Type	Mechanism of Action	References
PD-1 inhibitors	Pembrolizumab	Melanoma; Hodgkin lymphoma; hepatocellular cancer (HCC); renal cell carcinoma (RCC); Head and neck squamous cell carcinoma (HNSCC); non-small-cell lung cancer (NSCLC)	A humanized IgG4 monoclonal kappa-antibody that targets the PD-1 membrane protein on the surface of T-cells, blocking it. PD-1 plays a role in the negative regulation of the immune system, preventing the activation of T-lymphocytes.	NCT02362594 NCT02362997 NCT02453594 NCT02684292 NCT03062358 NCT03142334 NCT03040999 NCT02504372 NCT02578680 NCT02775435
	Nivolumab	Melanoma; Non-Hodgkin lymphoma; Metastatic clear cell renal carcinoma; Head and neck cancer; Metastatic colorectal cancer; NSCLC		NCT01721772 NCT01844505 NCT03068455 NCT02181738 NCT03366272 NCT01668784 NCT03342352 NCT02060188 NCT02041533
PD-L1 inhibitors	Durvalumab	Recurrent squamous cell lung cancer; Squamous cell lung carcinoma; Urothelial cancer; NSCLC	It is aimed at blocking the membrane protein PD-L1, which is a ligand for the PD-1 receptor. PD-L1 can be expressed on the cell surface of a tumor cell. When the PD-1: PD-L1 complex is formed, the T-cell receptor (TCR)-mediated positive signal is inhibited, which leads to a decrease in the activity and proliferation of T-lymphocytes.	NCT02766335 NCT02154490 NCT02516241 NCT03003962 NCT01693562 NCT03164616
	Avelumab	RCC; Gastric and gastroesophageal junction; Urothelial cancer; NSCLC		NCT02684006 NCT02625623 NCT02603432 NCT02576574
	Atezolizumab	RCC; Bladder cancer; Urothelial carcinoma; Prostatic neoplasms; NSCLC		NCT03024996 NCT02302807 NCT02807636 NCT03016312 NCT03191786 NCT03456063
CTLA-4 inhibitors	Ipilimumab	Metastatic melanoma; Gastric cancer; NSCLC	It is aimed at blocking the receptor protein CTLA-4, which is a homologue of CD28 on the surface of the T-lymphocyte, exerting an inhibitory effect on it. When T-lymphocytes interact with antigen presenting cells (APC), CTLA-4 prevents the activation of T-cells, because it has a higher affinity for the costimulatory domains CD80 and CD86 compared to the CD28 receptor.	NCT03445533 NCT02872116 NCT03351361 NCT03469960
NKG2A inhibitors	Monalizumab	Recurrent or metastatic HNSCC; NSCLC	It is aimed at blocking the NKG2A/CD94 receptors found in NK cells. Tumor cells use HLA-E protein molecules to bind to NKG2A, thereby exerting an inhibitory effect on the cytotoxic activity of NK cells.	NCT02643550 NCT03822351 NCT03833440

**Table 2 biomedicines-08-00621-t002:** Comparative table of generations of chimeric antigen receptor (CAR) T-cell therapy.

CARs Generations	Domains	Efficiency, (%)	Cancer Type	References
I	scFv; CD3ζ	Absent	Ovarian cancer	[202]
II	scFv; CD28; CD3ζ	CR (11.5), PR (46), CRS (100)	Multiple myeloma	NCT02215967
	scFv; CD28; CD3z	CR (83), CRS (85), ICANS (43), median OS 12.9 months	Chemotherapy resistant or refractory ALL	NCT01044069
	scFv; 4-1BB; CD3ζ	CR (69), no response (31), CRS (94), ICANS (40), median OS 19.1 months		NCT01029366 NCT02030847
III	scFv; CD28; 4-1BB; CD3z	CR (40), CRS (20), median OS 6 months	B-cell lymphoma or leukemia	NCT02132624
	scFv; CD28; OX40; CD3ζ/CD3z, iCasp9	NR	Neuroblastoma	NCT01822652
	scFv; CD28; OX40; CD3z		Neuroblastoma, Osteosarcoma	NCT01953900
IV	scFv; CD28; NFAT promoter; CD3ζ/CD3z	NR	Metastatic melanoma	NCT01236573
	scFv; CD28; CD27; CD3z; 2A; iCasp9		PCNSL	NCT03125577

NFAT promoter—a nuclear factor of the activated T-cell responsive expression cassette for the inducible expression of a transgenic cytokine; iCasp9—iCaspase Suicide Safety Switch; CR—complete response; CRi—complete response with incomplete blood count recovery; NR—no results; CRS—cytokine release syndrome; ICANS—immune effector cell-associated neurotoxicity syndrome; OS—overall survival; ALL—acute lymphoblastic leukemia; PCNSL—primary central nervous system lymphoma.

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
