# Peer review of "Current Trends in Cancer Immunotherapy"

_biomedicines, 2020, doi:10.3390/biomedicines8120621_

Round 1
Reviewer 1 Report
Overall a comprehensive review of immunotherapy and provides a nice history of the progression of the field. Would recommend shortening some sections and make more concise. Also, some of the clinical data may be too specific and the reasoning for citing specific studies is unclear. Also in discussion of current trends of cancer immunotherapy, may spend less time on the monoclonal antibodies, etc. and more on other adoptive cell therapy methods (TILs, CAR macrophages, etc.)
Monoclonal antibody section: Would generally shorten descriptions. These are not as novel so don't think you need comprehensive assessment of different data. Would shorten and make more general. I do like how you separate it out by categories, though.
1) Bevacizumab has many indications. Not sure why these specific studies were cited. Would shorten this and just describe indications for bevacizumab.
2) Not sure if need so much data about pertuzumab/trastuzumab. Would also make more general and shorten (don't need 2 paragraphs on this).
3) Don't know if need to list specific study of rituximab. Could just generally site improved outcomes in lymphoma with rituximab and maybe briefly mention second generation anti-CD20 agents (obinutuzumab) and hwo they are different.
4) For antibody drug conjugates and discussion of brentuximab vedotin, I would remove the retrospective study described. There are multiple prospective randomized trials but may actually be better just to state that BV has shown a lot of promise and being moved to earlier lines of therapy. Then consider mentioning other antibody drug conjugates other ADCs in development.
5) I think hard to categorize all of the side effects for these in 1 paragraph and I think better to discuss them by class types (potentially focus on rituximab infusion reactions, immune side effects of bi-specific T cell engagers such as blinatumomab, etc.). Also would be helpful to have more of a discussion of mechanism of side effects.
6) Could add discussion about resistance/lack of response.
Immune Checkpoint Inhibitors
1) Would create a separate section for this.
2) Not sure why chose the particular studies to discuss ICIs. I don't think metastatic colorectal cancer is the best clinical example. Could present some melanoma data and then talk about lung cancer (and possibly Hodgkin lymphoma). Then discuss the reason behind response to checkpoint inhibitors and go into unique toxicities.
Vaccines
1) In peptide vaccine section, unclear which diseases referring to for these vaccine applications.
CAR-T:
1) Would describe CRS and neurotoxicity more and for neurotoxicity, the preferred term is now ICANS instead of CRES.
2) Table 2 describing CAR-T cell generation is a little misleading since there are so many different types of CART cells in each generation. Could consider including domains and diseases studied.
Author Response
Response to Reviewer 1 Comments
Overall a comprehensive review of immunotherapy and provides a nice history of the progression of the field. Would recommend shortening some sections and make more concise. Also, some of the clinical data may be too specific and the reasoning for citing specific studies is unclear. Also in discussion of current trends of cancer immunotherapy, may spend less time on the monoclonal antibodies, etc. and more on other adoptive cell therapy methods (TILs, CAR macrophages, etc.)
Point 1: Monoclonal antibody section: Would generally shorten descriptions. These are not as novel so don't think you need comprehensive assessment of different data. Would shorten and make more general. I do like how you separate it out by categories, though.
1) Bevacizumab has many indications. Not sure why these specific studies were cited. Would shorten this and just describe indications for bevacizumab.
Response: 1) We added more indications for bevaziumab and removed research description on lines 211-214.
“Bevacizumab, an FDA-approved angiogenesis inhibitor, is indicated for the treatment of metastatic colorectal cancer (mCRC) [40], non-small-cell lung cancer (NSCLC) [41], metastatic breast cancer (mBC) [42], glioblastoma multiforme (GBM) [43], renal cell carcinoma (RCC) [44], ovarian cancer (OC) [45] and cervical cancer (CC) [46].”
2) Not sure if need so much data about pertuzumab/trastuzumab. Would also make more general and shorten (don't need 2 paragraphs on this).
Response: 2) We left the general information on lines 215-234.
3) Don't know if need to list specific study of rituximab. Could just generally site improved outcomes in lymphoma with rituximab and maybe briefly mention second generation anti-CD20 agents (obinutuzumab) and hwo they are different.
Response: 3) We removed researches description and added information about obinutuzumab and their difference on lines 244-252.
“Clinical trials have shown that this drug not only prolongs PFS but also increases OS in patients with lymphoma [55]. But development of resistance in rituximab-treated patients was observed [56]. Mechanisms of resistance were explained by trogocytosis of mAb-CD20 complexes [57] and by internalization of rituximab from the surface of the B-cells malignancies [58]. Obinutuzumab is a glycoengineered, humanized anti-CD20 mAb (type II) that has been developed to increase activity by enhancing binding affinity to the FcγRIII receptor on immune effector cells and due to the ability of enhancing direct cell death and antibody-dependent cell-mediated cytotoxicity/antibody-dependent cellular phagocytosis because of the modified elbow-hinge amino acid sequence [59].”
4) For antibody drug conjugates and discussion of brentuximab vedotin, I would remove the retrospective study described. There are multiple prospective randomized trials but may actually be better just to state that BV has shown a lot of promise and being moved to earlier lines of therapy. Then consider mentioning other antibody drug conjugates other ADCs in development.
Response: 4) We removed retrospective study and added general information about BV and other ADCs on lines 259-268.
“This drug has shown its safety and efficacy in many clinical trials, and is being moved to earlier lines of therapy in ongoing trials [61]. Other antibody-drug conjugates (ADCs) that were approved by FDA and European Medicines Agency (EMA) are gemtuzumab ozogamicin for treating CD33-positive acute myeloid leukemia in combination and as a single-agent therapy [62,63]; trastuzumab emtansine for treating HER2-positive breast cancer for patients who previously received trastuzumab and a taxane [64,65]; inotuzumab ozogamicin for treating relapsed or refractory CD22-positive B-cell precursor ALL in combination and as a single agent [66]. There are also several ADCs that are under development and are being clinically tested such as Glembatumumab vedotin, Sacituzumab govitecan, Anetumab ravtansine, Coltuximab ravtansine, Trastuzumab deruxtecan and GSK2857916 [67].”
5) I think hard to categorize all of the side effects for these in 1 paragraph and I think better to discuss them by class types (potentially focus on rituximab infusion reactions, immune side effects of bi-specific T cell engagers such as blinatumomab, etc.). Also would be helpful to have more of a discussion of mechanism of side effects.
Response: 5) We added more information about side effects on lines 286-305.
“The most frequent side effects are infusion reactions. Such side effects can occur due to allergic reactions to foreign proteins or as a result of cytokine release. All described infusion reactions were observed in patients in the first infusions of rituximab, however they. varied in severity and included different symptoms [69]. Moreover, specific side effects may occur when mAbs targeting an antigen are used. Treatment with bevacizumab can cause gastrointestinal perforation, hypertension, bleeding, nausea, diarrhea, thromboembolic complications, and less commonly skin ulcers [70,71], in addition to bowel ischaemia and haemorrhage [72]. Conjugated mAbs can exhibit higher toxicity due to conjugated substances – toxic chemotherapy drugs or radionuclides. For example, in patients who have been treated with brentuximab vedotin, the most common side effects were peripheral sensory neuropathy (PSN), neutropenia, fatigue, nausea, anaemia, upper respiratory tract infection (URTI), diarrhoea, thrombocytopaenia and coughing [60]. More severe side effects are related to bispecific antibodies. Side effects of the central nervous system (CNS) such as encephalopathy, aphasia, tremor, disorientation, and seizure are observed in patients treated with blinatumomab [73]. In one study, 52% of patients in original cohort and 42% in the additional evaluation cohort had neurologic events (NEs). Moreover, 4 patients had grade 4 NEs (encephalopathy, ataxia, seizures and febrile delirium), and two of them were fatal after treatment had stopped [74]. The main reason for such side effects can be explained by adherence of blinatumomab-activated T-cells to the endothelium, so that activated T-cells transmigrate across the blood brain barrier and enter the CNS where they induce a T-cell mediated toxic inflammation of the CNS [73].”
6) Could add discussion about resistance/lack of response.
Response: 6) We added information about lack of response on lines 306-312.
“Some patients who have been treated with mAbs then show resistance to this therapy. It has been noticed that a high percentage of Treg and high serum lactate dehydrogenase levels are related to lack of response to blinatumomab. It is supposed that activation of Treg by blinatumomab, and further IL-10 production, results in suppression of T-cell proliferation [75]. In addition lack of response can be as a consequence of the formation antibody-drug antibodies (ADAs). What is more, patients on infliximab who have developed ADAs also developed ADAs to adalimumab [76]. Thus it is necessary to calculate the dose of the drug to minimize the likelihood of ADAs development.”
Point 2: Immune Checkpoint Inhibitors
1) Would create a separate section for this.
Response: 1) We created a separate section on line 313.
2) Not sure why chose the particular studies to discuss ICIs. I don't think metastatic colorectal cancer is the best clinical example. Could present some melanoma data and then talk about lung cancer (and possibly Hodgkin lymphoma). Then discuss the reason behind response to checkpoint inhibitors and go into unique toxicities.
Response: 2) We removed all mCRC clinical trials and added melanoma and Hodgkin lymphoma trials on lines 332-356.
“Nivolumab and dacarbazine were compared in patients with previously untreated BRAF wild-type advanced melanoma. At a minimum follow-up of 38.4 months among 210 participants (nivolumab group) and 208 participants (dacarbazine group) the median OS was 37.5 months and 11.2 months, respectively. CR and PR were 19.0% and 23.8% in the nivolumab group compared with 1.4% and 13.0% in the in the dacarbazine group [86]. Nivolumab in combination with ipilimumab showed better results in patients with advanced melanoma than nivolumab alone or ipilimumab alone. Overall survival at 5 years was 52% in the nivolumab-plus-ipilimumab group and 44% in the nivolumab group, as compared with 26% in the ipilimumab group [87]. With pembrolizumab plus chemotherapy for NSCLC, the median OS was 15.9 months, compared with chemotherapy alone, where the median OS was 11.3 months [88]. In Phase I/II clinical trials, the efficacy of durvalumab in patients with NSCLC was assessed, having previously determined the level of PD-L1 expression in tumor cells, considering the result to be positive at a PD-L1 expression level of ≥25%. ORR was 15.3% of the total number of patients, where 21.8% in patients with PD-L1 expression ≥25% and 6.4% in patients with PD-L1 expression <25%. The median OS of the total number of patients was 12.4 months, which broke down to 16.4 months for PD-L1 ≥25% and 7.6 months for PD-L1 <25%. The median PFS overall was 1.7 months, with 2.6 and 1.4 months in patients with PD-L1 expression ≥25% and <25%, respectively. Moreover, side effects were observed in 10.2% of patients [89]. The efficacy of pembrolizumab in patients with relapsed or refractory Hodgkin lymphoma was also analyzed. With a median follow-up of 27.6 months ORR was 71.9%, the CR was 27.6%, and the PR rate was 44.3% [90]. In another study pembrolizumab in patients with relapsed/refractory classical Hodgkin lymphoma after autologous stem cell transplantation showed promising results. The PFS at 18 months for the 28 evaluable patients was 82%. The 18-month overall survival was 100% [91]. Nivolumab for newly diagnosed advanced-stage classic Hodgkin lymphoma was associated with promising efficacy. The ORR was 84%, 67% achieve complete remission. With a minimum follow-up of 9.4 months the PFS was 92% [92].”
Also we added more information about toxicity on lines 362-369 and 381-389.
“During therapy, irAEs of various natures and genesis can be observed, including several organs or systems, for example: skin diseases (itching, rash, eczema, vitiligo) with 30-50% frequency; thyroid dysfunction (hypothyroidism, hyperthyroidism and thyroiditis) in 6-20% of patients; hypophysitis in 1-7% of patients; gastrointestinal diseases like diarrhea occurs in one third of patients or colitis in 8-22%; liver disease (hepatitis) with 1-4% frequency, but its reached 17.6%; lung diseases such as pneumonitis with 5-10% frequency and the most common dyspnoea (53%) and cough (35%); rare endocrine adverse events such as primary adrenal insufficiency (0.7%) and insulin-deficient diabetes (0.2%) and others [93]. The risk of most iAEs is dose-dependent.”
“The development of irAEs is commonly associated with inflammation and autoimmune reactions due to cross-reactivity of antibodies and cytokines. Such cross-reactivity takes place because of the antigenic resemblance between tumor cells and host cells [96]. It is hard to define the frequency and severity of irAEs because it depends on a variety factors such as dose of ICIs, their mechanism of action and combination with other therapeutic agents, other features of patients (underlying autoimmune disease, organ or hematopoietic stem-cell transplants, chronic viral infection, organ dysfunction, or advanced age) [96]. Nowadays, clinical practice guidelines are available that can help to manage with irAEs of different origins and severity [97].”
Point 3: Vaccines
1) In peptide vaccine section, unclear which diseases referring to for these vaccine applications.
Response: 1) Added information on lines 471-473.
“A lot of different peptide-based vaccines are used for the treatment of numerous types of cancers such as glioma [123], breast cancer [124], hematopoietic tumors [125], renal cell carcinoma [126] and others.”
Point 4: CAR-T:
1) Would describe CRS and neurotoxicity more and for neurotoxicity, the preferred term is now ICANS instead of CRES.
Response: 1) We reversed the term for neurotoxicity and added more information about CRS and ICANS on lines 840-855.
“The first FDA approved drug involving CAR T-cells was tisagenlecleucel (Kymriah) in ALL patients up to 25 years of age in 2017 [218]. Kymriah helped to reach complete remission in ~60-90% of patients with refractory or relapsed ALL. However, there is still a high risk of developing CRS and ICANS. Fever, rigors, tachycardia, hypotension, tachypnea, hypoxemia, and other signs of systemic inflammation are common observed in CRS. Such CRS are reversible in most patients, but there are also grade ≥3 or 4 CRS, sometimes with fatal cases, that have been reported. Severe grade ≥3 CRS was characterized by profound hemodynamic instability, capillary leak syndrome, and consumptive coagulopathy. Developing of ICANS is more complicated. Neurological symptoms such as headache, tremor, speech impairment, delirium, confusion, impaired consciousness have been observed [219]. Endothelial damage, microglial activation, parenchymal necrosis, multifocal microhemorrhages and CAR T-cell infiltration of the brain in patients with fatal cerebral edema after CD19 CAR T cell therapy have been reported. However, severe ICANS is rarely observed in the absence of severe CRS [220]. It is important to count thoroughly the CAR T-cell dose to prevent severe CRS and ICANS. Moreover, administration of corticosteroids, tocilizumab (IL-6 receptor-directed antibody) and anakinra (IL-1 receptor antagonist) may reduce CRS and ICANS severity [219].”
2) Table 2 describing CAR-T cell generation is a little misleading since there are so many different types of CART cells in each generation. Could consider including domains and diseases studied.
Response: 2) Added more information in Table 2. (Please see the attachment)
Reviewer 2 Report
In this review article, Filin et al. review “current trends in immunotherapy” (also the title of their article) focusing on cytokines, monoclonal antibodies, vaccines, oncolytic viruses, and chimeric antigen receptors as treatment modalities for cancer. While subsections on some of these therapies are well written, the introduction section is written very incoherently, with insufficient and sometimes inaccurate explanations. For example, TSAs are referred to as chimeric proteins. The introduction will benefit from scientifically accurate and clear explanations, as review articles often serve as a great resource for new researchers to develop their grasp about the subject, and to quickly understand what has already been accomplished in the field. Additionally, at several places in the review overall, immunologically relevant phenomena are described with non-scientific terms trivializing the complexity of these processes.
Other comments are:
Line 32: “suppressing” MHC molecules is not accurate. The authors may want to mention downregulation of MHC expression, which in turn results in lack of tumor recognition by CTLs.
Line 41: It is not clear what the “factors” are in line 41.
Line 63: “Immediate capture” of neoantigens is not accurate. Phagocytose/endocytose is scientifically accurate when describing antigen uptake by dendritic cells.
Neoantigens can be described alongside TAAs, TSAs in the introduction.
Line 66: “Determination” processes: unclear what this is referring to.
Line 94: It is unclear when authors mention “antigens as their own, and not foreign” and how this impacts immune cells in redirecting toward a Treg response
Line 98: This sentence seems like an oversimplification of tumor immunotherapy, where authors suggest to “reactivate all steps involved”. Like any other biological phenomenon, authors must realize anti-tumor response is a complex process, and it seems highly unlikely that all that is described in Figure 1 will be targeted by a single treatment.
At several places throughout the review, IFN-gamma is mentioned as INF-gamma.
Lines 200-203: adalimumab, muromonab, blockade: incorrect spelllings.
Lines 226: When describing combination treatment with trastuzumab, it is worth mentioning pertuzumab also targets HER2.
Lines 246-247: Transition from solid tumor treatment to hematologic tumors is not smooth. Some description of how the two tumor types differ will help. Also, describing CD20 is an antigen on B cells that is being targeted by rituximab in these malignancies will help.
Lines 338-339: It is confusing what the authors refer to when they mention "safest method of administering". Are they talking about intraperitoneal, intravenous .. or other such method of administration?
Line 342: It appears something is missing here when mentioning the 10 mg/kg dose.
Lines 346-348: Immunotherapy has clearly been one of the most successful methods for treating cancer in the past 2 decades. The adverse events remain a bottleneck that needs to be resolved. Authors should credit the therapy appropriately, in addition to describing the adverse events. Authors can also describe possible reasons for why the adverse events may occur – cross reactivity with a related antigen for example?
Line 354: Not clear which “therapy” the authors refer to here.
Lines 357-362: it is not clear how the TMEs were classified as immune desert, immune excluded or inflamed. Appropriately referencing, Chen and Mellman (ref 74), the authors can state that based on histology studies of patient tumors, Chen and Mellman describe three phenotypes that can have an effect on the immune response generated upon anti-cancer treatment (in this case, anti-PD-L1/PD-1 therapy), and then describe the three profiles.
Line 363: “Effective strategies directed at overcoming these resistance mechanisms need to be personalized first.”: This standalone statement does not seem to fit in without further explanation.
Line 380” Instead of using the word “connection”, the authors may want to use a more descriptive word to provide information to the reader. For example, “correlation”.
Section 5, Vaccines: Before talking about cancer vaccines, it may be useful to describe to the reader about conventional vaccines (in context with infectious diseases), and then following up about the concept of cancer vaccines, and how they can be used against cancer. Somewhere in this section, the authors may also want to talk about neoantigen peptide vaccines (example: Ott et al, Nature, 2017, doi:10.1038/nature22991)
Line 426: “limitation of HLA” is not accurate. The authors may want to attribute the unsuitability of the peptide vaccines to the diversity in human MHC alleles (HLA), and the different requirements of each allele in binding to the peptide.
Line 427: “Optimized” peptides sounds vague. What needs to be optimized?
Line 477: The authors can describe in a few words how peptides, proteins, tumor cells, or lysates can be used as DC vaccine, not just direct the reader to more reviews.
Lines 536-537: “Many different clinical trials are underway around the world of which we can confidently say that DCs-based vaccines are safe and non-toxic.”: I wouldn't recommend making such statements. Instead, put in data/numbers to show efficacy to the reader. You can say a trial is "promising” but cannot "confidently" say it is the best.
Line 672: “much more simply”: phrase is ambiguous.
Line 676: The interaction of the TCR and peptide-MHC, not MHC.
Line 678: I do not believe, “improving” potential of TCRs is accurate. Recognizing antigen without MHC as a chimeric antigen receptor, allows T cells to target antigens that are targeted by antibodies (scFv in CAR), but T cells provide an additional benefit of potency via their signaling, when targeting an antigen.
Line 686-687: This was clearly a breakthrough technology, and hence it is probably not appropriate to say that first generation CARs were “unsuccessful” while referring to the initial breakthrough papers. The efficacy was low, but clearly the first generation CARs were the stepping stone to generate the second generation CARs and so on..
Line 692: Second generation CAR: No “other things” were included in second generation CAR, except for the domain from costimulatory molecules, in addition to the scFv and CD3z domain which was already present in the first generation CAR.
Lines 802-803: Loss of antigen (CD19) is well recognized in the CAR field. The authors can describe targeting multiple antigens (example CD19 and CD20 targeting CAR studies) as a strategy to tackle antigen loss by the cancer.
Author Response
Response to Reviewer 2 Comments
In this review article, Filin et al. review “current trends in immunotherapy” (also the title of their article) focusing on cytokines, monoclonal antibodies, vaccines, oncolytic viruses, and chimeric antigen receptors as treatment modalities for cancer. While subsections on some of these therapies are well written, the introduction section is written very incoherently, with insufficient and sometimes inaccurate explanations. For example, TSAs are referred to as chimeric proteins. The introduction will benefit from scientifically accurate and clear explanations, as review articles often serve as a great resource for new researchers to develop their grasp about the subject, and to quickly understand what has already been accomplished in the field. Additionally, at several places in the review overall, immunologically relevant phenomena are described with non-scientific terms trivializing the complexity of these processes.
Response: Changed the explanation of the TSAs (Line 40).
“Tumor-specific antigens (TSAs) are fragments of novel peptides that are presented by MHC-I at the cell surface.”
Line 32: “suppressing” MHC molecules is not accurate. The authors may want to mention downregulation of MHC expression, which in turn results in lack of tumor recognition by CTLs.
Response: “suppressing” was replaced by “downregulate” (Line 33)
Line 41: It is not clear what the “factors” are in line 41.
Response: The word “factors” was replaced by “presence of such antigens” (Line 44)
Line 63: “Immediate capture” of neoantigens is not accurate. Phagocytose/endocytose is scientifically accurate when describing antigen uptake by dendritic cells.
Response: “immediate react and capture them” was replaced by “take up them by phagocytosis or endocytosis” (Line 66)
Neoantigens can be described alongside TAAs, TSAs in the introduction.
Response: Added information about neoantigens on lines 43-44.
“There are also neoantigens that are the result of somatic mutations and are specific to each patient, differing from wild-type antigens.”
Line 66: “Determination” processes: unclear what this is referring to.
Response: The word “Determination” replaced by “differentiation” (Line 69)
Line 94: It is unclear when authors mention “antigens as their own, and not foreign” and how this impacts immune cells in redirecting toward a Treg response
Response: Changed this sentence on: “Immune cells can also create favorable conditions within the tumor microenvironment (TME), in particular, Treg have an immunosuppressive effect on CTLs by anti-inflammatory cytokines (IL-10 and TGF-β), forcing them to differentiate into Treg.” (Lines 96-99)
Line 98: This sentence seems like an oversimplification of tumor immunotherapy, where authors suggest to “reactivate all steps involved”. Like any other biological phenomenon, authors must realize anti-tumor response is a complex process, and it seems highly unlikely that all that is described in Figure 1 will be targeted by a single treatment.
Response: Agree, changed sentence on: “Under these conditions, antitumor immunotherapy should be aimed at enhancing immune responses and preventing immunosuppression.” (Lines 101-103)
At several places throughout the review, IFN-gamma is mentioned as INF-gamma.
Response: Corrected
Lines 200-203: adalimumab, muromonab, blockade: incorrect spelllings.
Response: Corrected
Lines 226: When describing combination treatment with trastuzumab, it is worth mentioning pertuzumab also targets HER2.
Response: Added information about pertuzumab and difference between trastuzumab and pertuzumab (Lines 217-220)
“Pertuzumab is another recombinant anti-HER2 humanized monoclonal antibody [49]. The difference is that trastuzumab and pertuzumab bind to different domains of HER2, which gives a synergistic effect [50].”
Lines 246-247: Transition from solid tumor treatment to hematologic tumors is not smooth. Some description of how the two tumor types differ will help. Also, describing CD20 is an antigen on B cells that is being targeted by rituximab in these malignancies will help.
Response: Added more information about hematologic tumors: “In addition to solid tumors, a lot of attention is also paid to hematological malignancies. Hematological malignancies, unlike solid tumors, begin in the blood-forming tissue of the bone marrow and lymphoid cells. Hematologic B-cell tumors represent a large heterogeneous group of lymphoproliferative disorders, including diseases such as follicular lymphoma (FL), chronic lymphocytic leukemia (CLL), mantle cell lymphoma (MCL), diffuse large B-cell lymphoma (DLBCL) and others [53,54]. B-cell transmembrane protein (CD20) was chosen for targeted therapy as it is expressed on most B-cells, including malignant B-cells [55].” (Lines 235-241)
Lines 338-339: It is confusing what the authors refer to when they mention "safest method of administering". Are they talking about intraperitoneal, intravenous .. or other such method of administration?
Response: Agree, it sounds misleading. Changed "safest method of administering" on “the safest dosage and frequency of administration” (Line 372-373)
Line 342: It appears something is missing here when mentioning the 10 mg/kg dose.
Response: Added clarification (3 milligrams of drug per kilogram of body weight) on lines 375-376
Lines 346-348: Immunotherapy has clearly been one of the most successful methods for treating cancer in the past 2 decades. The adverse events remain a bottleneck that needs to be resolved. Authors should credit the therapy appropriately, in addition to describing the adverse events. Authors can also describe possible reasons for why the adverse events may occur – cross reactivity with a related antigen for example?
Response: Added more information about adverse events on lines 381-389. “The development of irAEs is commonly associated with inflammation and autoimmune reactions due to cross-reactivity of antibodies and cytokines. Such cross-reactivity takes place because of the antigenic resemblance between tumor cells and host cells [96]. It is hard to define the frequency and severity of irAEs because it depends on a variety factors such as dose of ICIs, their mechanism of action and combination with other therapeutic agents, other features of patients (underlying autoimmune disease, organ or hematopoietic stem-cell transplants, chronic viral infection, organ dysfunction, or advanced age) [96]. Nowadays, clinical practice guidelines are available that can help to manage with irAEs of different origins and severity”
Line 354: Not clear which “therapy” the authors refer to here.
Response: Added clarification: “ICIs therapy” (Line 395)
Lines 357-362: it is not clear how the TMEs were classified as immune desert, immune excluded or inflamed. Appropriately referencing, Chen and Mellman (ref 74), the authors can state that based on histology studies of patient tumors, Chen and Mellman describe three phenotypes that can have an effect on the immune response generated upon anti-cancer treatment (in this case, anti-PD-L1/PD-1 therapy), and then describe the three profiles.
Response: Agree, added information “Daniel Chen and Ira Mellman distinguished three basic immune profiles that correlate with immune response to anti-PD-L1/PD-1 therapy, while they examined histological studies of patient tumors [102].” (Lines 398-400)
Line 363: “Effective strategies directed at overcoming these resistance mechanisms need to be personalized first.”: This standalone statement does not seem to fit in without further explanation.
Response: Agree, changed the sentences in places to keep the meaning: “Effective strategies directed at overcoming these resistance mechanisms need to be personalized first. Currently, special prognostic markers are actively being developed which could determine the outcome of a therapy before starting it, which can dramatically help in solving this problem by increasing the effectiveness of therapies and reducing the risk of irAEs [103,104].”. (Lines 405-409)
Line 380” Instead of using the word “connection”, the authors may want to use a more descriptive word to provide information to the reader. For example, “correlation”.
Response: The word “connection” was replaced by “correlation” on line 421
Section 5, Vaccines: Before talking about cancer vaccines, it may be useful to describe to the reader about conventional vaccines (in context with infectious diseases), and then following up about the concept of cancer vaccines, and how they can be used against cancer. Somewhere in this section, the authors may also want to talk about neoantigen peptide vaccines (example: Ott et al, Nature, 2017, doi:10.1038/nature22991)
Response: 1) Added information about conventional vaccines on lines 433-439. “Nowadays vaccines are the most effective treatment against infectious diseases. We have managed to prevent such diseases as smallpox, yellow fever, rubella, polio and measles [111]. There are two types of vaccines: prophylactic and therapeutic. Both types of vaccines aim to elicit specific immune responses. Preventive vaccines act against pathogenic microorganisms or oncogenic viruses (for example, human papillomavirus, HPV) based on attenuated or killed pathogens or virus-like particles (VLP). Therapeutic vaccines are based, for example, on autologous human immune cells or peptides to fight tumor cells.”
2) Added more information about neoantigen peptide vaccines on lines 487-493. “On the other hand, vaccines based on neoantigens have unique peptide sequences, these are more personalized and hold lower risks for autoimmune generation which makes them promising targets for activating immune responses [131]. The Dana Farber Cancer Institute (DFCI) recently published promising results in the NeoVax trial in which 4 of 6 patients who were treated with long synthetic neoantigens peptides plus poly IC:LC adjuvant had no recurrence at 25 months following treatment. Wherein 2 patients were additionally treated with anti-PD-1 inhibitors and experienced complete tumor regression [132].”
Line 426: “limitation of HLA” is not accurate. The authors may want to attribute the unsuitability of the peptide vaccines to the diversity in human MHC alleles (HLA), and the different requirements of each allele in binding to the peptide.
Response: Agree, we described limitation of peptide vaccines more accurate: “These vaccines are not suitable for every patient, due to the diversity in human MHC alleles, therefore peptides with lower affinities for the MHC may be less immunogenic and cannot be presented for the circulating naïve T-cells on ongoing basis. Another limitation lies in rapid degradation of short peptides by serum and tissue peptidases [128].” (Lines 475-478)
Line 427: “Optimized” peptides sounds vague. What needs to be optimized?
Response: Agree, changed the wording in the proposal to “Specially improving peptide based vaccines for adaptation to a large group of patients like modification of adjuvants, new immunogenic neoantigens and combination with another immunotherapeutic agents may help avoid these problems [128].” (Lines 478-481)
Line 477: The authors can describe in a few words how peptides, proteins, tumor cells, or lysates can be used as DC vaccine, not just direct the reader to more reviews.
Response: Added description of DC loading on lines 537-546. “Tumor cells need to inactivate by 24-hour TNF sensitization, gamma irradiation and 48-hour cultivation in serum-free medium. After that DCs cultivate with inactivate tumor cells for a few hours [145]. Tumor lysate is obtained by four freeze (-80°Ð¡) and thaw (room temperature) cycles. Then DCs cultivate with tumor lysate overnight [146]. Peptides are synthesized by automated solid phase synthesis and cultivate with mature DCs for 4-6 hours [147]. Alternative methods are viral vectors and transfection of DNA and mRNA encoding TAAs for presentation and induced maturation of DCs by lentiviral transduction and electroporation, respectively [148-150]. In phase IB clinical trial DCs electroporated with mRNA encoding proteins for maturation and with mRNA encoding fusion proteins of a HLA-class II targeting signal and a melanoma-associated antigen [151].”
Lines 536-537: “Many different clinical trials are underway around the world of which we can confidently say that DCs-based vaccines are safe and non-toxic.”: I wouldn't recommend making such statements. Instead, put in data/numbers to show efficacy to the reader. You can say a trial is "promising” but cannot "confidently" say it is the best.
Response: Agree, changed the wording in the proposal to “Many different clinical trials are underway around the world in which DCs-based vaccines show their safety and efficacy in combination with other immunotherapy agents (NCT01302496, NCT01876212, NCT03152565, NCT00266110).” (Lines 602-604)
Line 672: “much more simply”: phrase is ambiguous.
Response: Changed “much more simply” on “simpler” (Line 739)
Line 676: The interaction of the TCR and peptide-MHC, not MHC.
Response: Changed “MHC” on “peptide-MHC” (Line 743)
Line 678: I do not believe, “improving” potential of TCRs is accurate. Recognizing antigen without MHC as a chimeric antigen receptor, allows T cells to target antigens that are targeted by antibodies (scFv in CAR), but T cells provide an additional benefit of potency via their signaling, when targeting an antigen.
Response: Agree, changed the wording in the proposal to “In order for TCR to recognize antigens without the participation of MHC and react on them, therefore it was proposed that combining the antigen-binding variable domain of antibodies with the constant domain of TCR in one polypeptide may be beneficial, due to the similarity in the structure and organization of TCR and immunoglobulin [197].” (Line 744-748)
Line 686-687: This was clearly a breakthrough technology, and hence it is probably not appropriate to say that first generation CARs were “unsuccessful” while referring to the initial breakthrough papers. The efficacy was low, but clearly the first generation CARs were the stepping stone to generate the second generation CARs and so on..
Response: Agree, changed on “The efficacy of the first experiments with the participation of such structures was low [198-201].” (Lines 752-753)
Line 692: Second generation CAR: No “other things” were included in second generation CAR, except for the domain from costimulatory molecules, in addition to the scFv and CD3z domain which was already present in the first generation CAR.
Response: Agree, changed the wording in the proposal to “Thus, the 2nd generation of CAR appeared, containing one costimulatory domain in addition to the scFv and CD3z domain.” (Lines 758-759)
Lines 802-803: Loss of antigen (CD19) is well recognized in the CAR field. The authors can describe targeting multiple antigens (example CD19 and CD20 targeting CAR studies) as a strategy to tackle antigen loss by the cancer.
Response: Agree, added information about loss of antigen on lines 887-894. “In order to overcome the loss of the target antigen, the CAR molecule was engineered to recognize multiple antigens by linking two binders on a single molecule (tandem CAR) [232]. Such bispecific CARs as CD19-CD20 or CD19-CD22 can increase the efficacy of the CAR T-cell therapy [233,234]. In 1 phase trial patients with resistance to CD19-targeted immunotherapy received CD19-CD22 CAR T-cells. Complete remission was obtained in 73% of patients and the median remission duration was 6 months [234]. The crucial role of antigen density in the regulation of receptor function was also assessed, since relapses were associated with a decrease in the density of CD22 sites.”
Reviewer 3 Report
This manuscript thoroughly reviewed the history and progress of cancer immunotherapy. It starts with a brief introduction to cancer immunity and extends to cytokines, monoclonal antibodies, anti-tumor vaccines, oncolytic viruses, and CAR Т-cell therapy. It covered most of the major milestones in the field.
Author Response
In this article, adjustments were made to the sections on cancer immunity, monoclonal antibodies, vaccines (peptide and dendritic), CAR T-cell therapy.
We also added information in the 1st and 2nd tables.
Round 2
Reviewer 1 Report
The authors have spent time revising the manuscript and it is improved.
Author Response
We have edited the introduction and immune checkpoint inhibitors sections.
Reviewer 2 Report
As I also mentioned in my previous review, I believe the Introduction section is incoherent. The authors start with T cell mediated immune response toward class I cancer antigens, followed by one or two sentences about TAAs, TSAs, neoantigens, etc. without an opening sentence that leads the reader into the classification of antigens. The authors abruptly close this section by mentioning that “presence of cancer antigens” is not enough for destruction of tumor. This is a very crude way of talking about reduced recognition of tumor antigens by T cells in an immunosuppressive cancer microenvironment. The T cells themselves are very potent, and this seems to have dampened in the introduction section. The authors then move on to briefly discussing cancer immunoediting (again, without leading into the reasoning for why they are discussing this). Finally, there is no introduction to cancer antigens recognized by antibodies in the Introduction. This would have been ok if the focus of the review was only T cell-mediated immune response toward cancer. The authors have a section on antibodies, and hence antibody-mediated recognition of cancer antigens can also be briefly introduced in this section.
Line 66: “take them up” is more appropriate
Line 98-99: I am not aware if CTLs (i.e. CD8 T cells) can be differentiated into Tregs (CD4 T cells). The authors may want to confirm this, and appropriately reference the study.
Lines 381-384: "The development of iRAEs.....and host cells". Cross-reactivity of cytokines is inaccurate. According to the reference cited (ref 96), irAEs occur because the activity of immune system is increased by ICIs. This may involve a more pro-inflammatory cytokine profile, but I do not believe the cytokines "cross-react". Similarly, I am unsure about the antibody cross-reactivity in the context of treatment with ICIs. Based on this review (ref 96), it appears the authors suggest there may be a role of autoantibodies and autoreactive T cells in causing adverse events, and I would suggest Filin et al. to do the same, and not make a conclusive statement about cross-reactivity of the antibodies (here, ICIs, i.e. anti-PD1 anti-CTLA4 antibody). In the specific case of ICIs treated melanoma, the review cited by the authors suggests that cross-reactivity of (the now activated) T cells could be at play causing vitiligo, because the T cells against tumor (melanoma) can also act against related antigen in normal cells. I would accordingly discuss this. It appears that why iRAEs occur in response to ICIs is not sufficiently clear, and I will suggest writing about it accordingly.
Author Response
As I also mentioned in my previous review, I believe the Introduction section is incoherent. The authors start with T cell mediated immune response toward class I cancer antigens, followed by one or two sentences about TAAs, TSAs, neoantigens, etc. without an opening sentence that leads the reader into the classification of antigens. The authors abruptly close this section by mentioning that “presence of cancer antigens” is not enough for destruction of tumor. This is a very crude way of talking about reduced recognition of tumor antigens by T cells in an immunosuppressive cancer microenvironment. The T cells themselves are very potent, and this seems to have dampened in the introduction section. The authors then move on to briefly discussing cancer immunoediting (again, without leading into the reasoning for why they are discussing this). Finally, there is no introduction to cancer antigens recognized by antibodies in the Introduction. This would have been ok if the focus of the review was only T cell-mediated immune response toward cancer. The authors have a section on antibodies, and hence antibody-mediated recognition of cancer antigens can also be briefly introduced in this section.
Response: Agree, we've edited the introduction section on the lines 30-68: “In 1909, Paul Ehrlich predicted that the immune system normally prevents the formation of carcinomas from various origins. The first demonstration of a specific immune system response was made almost half a century later in 1953 by Gross [1]. Unfortunately, malignant neoplasms may evade such immune responses. It is known that malignant neoplasms downregulate the molecules of the major histocompatibility complex (MHC)-I, thereby preventing recognition of tumor cells by cytotoxic T-lymphocytes (CTLs). However, the immune system, in turn, is able to destroy cells that do not express, or insufficiently express, MHC-I molecules on their surface using natural killer cells (NK-cells) [2]. But tumor cells can also protect themselves from NK-cell lysis by expressing non-classic human leukocyte antigen (HLA)-G molecules on their surface [3]. Additionally, tumor cells can trigger angiogenesis [4], and can also recruit T-regulatory cells with immunosuppressive properties via chemical signals [3]. Thus, malignant neoplasms may evade the host immune response creating a tumor microenvironment. Gavin Dunn and Robert Schreiber developed the concept of "cancer immunoediting" in three phases. In the first phase, tumor cells are eliminated by cells of the immune system (NK cells, CD4+ and CD8+ T-lymphocytes) [5]. In the second phase, there is an equilibrium between tumor cells and cells of the immune system. In the third phase, the immune system is unable to cope with the tumor, which has an impressive immunosuppressive effect, therefore the phase ends with the appearance of a clinically detectable tumor [5]. Nowadays, the priority task is to create an effective immunotherapeutic method with minimal toxicity to overcome the immunosuppressive activity of the tumor cells and enhance targeted elimination of the tumor by the immune system host cells. Most immunotherapeutic methods, for example monoclonal antibodies, target specific antigens on the tumor cell surface to produce effective and accurate actions, and some methods, like dendritic vaccines, use these antigens to enhance the immunostimulating and immunomodulatory immune system activity. There are some types of antigens located on the surface of tumor cells that can induce a specific immune response. Such antigens were shown by Richmond Prehn and Joan Mine in their murine experiments in 1957 [6]. Subsequently, the so-called tumor-associated antigens (TAAs) were discovered. These include molecules that are expressed on the surface of cells in a prevailing amount, or in a state different from that observed in normal cells. Other tumor biomarkers include tumor-specific antigens (TSAs), which are fragments of novel peptides that are presented by MHC-I at the cell surface. The first TSAs were discovered in 1991 in human melanoma cells, which are encoded by the melanoma-associated antigen (MAGE) gene family [7]. There are also neoantigens that are the result of somatic mutations and are specific to each patient, thus differing from wild-type antigens. These antigens are used in immunotherapy methods like a target for recognition and subsequent elimination of tumor cells. Therefore, cancer immunotherapy aims to use the memory and specificity of the immune system to effectively eliminate malignant neoplasms for long periods of time and with minimal toxicity. Immunotherapy methods are also aimed at stimulating the body's own immune system in order to fight off the tumor. Immunotherapeutic methods presently include cytokine therapy, monoclonal antibodies, oncolytic viruses, prophylactic and therapeutic vaccines, and chimeric antigen receptor (CAR) T-cell therapy.”
Line 66: “take them up” is more appropriate
Response: Corrected to “take them up” on line 75.
Line 98-99: I am not aware if CTLs (i.e. CD8 T cells) can be differentiated into Tregs (CD4 T cells). The authors may want to confirm this, and appropriately reference the study.
Response: Agree, we changed the wording in the proposal to “Immune cells can also create favorable conditions within the tumor microenvironment (TME), in particular, Treg have an immunosuppressive effect on DCs and CTLs by anti-inflammatory cytokines (IL-10 and TGF-β). Moreover, these cytokines force CD4+ T-cells to differentiate into Treg which have suppressive properties for both Th1 and Th2 [3].” (Lines 105-109)
Lines 381-384: "The development of iRAEs.....and host cells". Cross-reactivity of cytokines is inaccurate. According to the reference cited (ref 96), irAEs occur because the activity of immune system is increased by ICIs. This may involve a more pro-inflammatory cytokine profile, but I do not believe the cytokines "cross-react". Similarly, I am unsure about the antibody cross-reactivity in the context of treatment with ICIs. Based on this review (ref 96), it appears the authors suggest there may be a role of autoantibodies and autoreactive T cells in causing adverse events, and I would suggest Filin et al. to do the same, and not make a conclusive statement about cross-reactivity of the antibodies (here, ICIs, i.e. anti-PD1 anti-CTLA4 antibody). In the specific case of ICIs treated melanoma, the review cited by the authors suggests that cross-reactivity of (the now activated) T cells could be at play causing vitiligo, because the T cells against tumor (melanoma) can also act against related antigen in normal cells. I would accordingly discuss this. It appears that why iRAEs occur in response to ICIs is not sufficiently clear, and I will suggest writing about it accordingly.
Response: The wording in proposal is incorrect. We changed the sentence on: “It is assumed that the development of irAEs is commonly associated with inflammation due to pro-inflammatory cytokines and autoimmune reactions due to the cross-reactivity of T-cells. Such cross-reactivity may occur because of the antigenic resemblance between tumor cells and host cells [98]. It is hard to define the frequency and severity of irAEs because it depends on a variety factors such as the dose of ICIs, their mechanism of action and combination with other therapeutic agents, and other features of individual patients (such as underlying autoimmune disease, organ or hematopoietic stem-cell transplants, chronic viral infection, organ dysfunction, or advanced age) [98]. It is known that vitiligo can develop in patients with melanoma who have undergone immune stimulation [98]. Since vitiligo is not a common irAE in patients with other cancers, it can be assumed that the development of vitiligo in melanoma patients treated with ICIs may be due to the cross-reactivity of T-cells. However, the cause of irAEs in response to ICIs is still not sufficiently understood.” (Lines 392-402)